# Macrophage-derived GPNMB trapped by fibrotic extracellular matrix promotes pulmonary fibrosis

Jing Wang[1], Xinxin Zhang[1], Min Long[2], Mengqin Yuan[2], Juan Yin[1,3], Wei Luo[1,3], Sha Wang[1], Yu Cai[4], Wei Jiang [2,6✉] & Jie Chao [1,3,5,6✉]

Pulmonary fibrosis (PF) is a form of progressive lung disease characterized by chronic inflammation and excessive extracellular matrix (ECM) deposition. However, the protein changes in fibrotic ECM during PF and their contribution to fibrosis progression are unclear. Here we show that changes in expression of ECM components and ECM remodeling had occurred in silica-instilled mice. The macrophage-derived glycoprotein nonmetastatic melanoma protein B (GPNMB) captured by fibrotic ECM may activate resident normal fibroblasts around the fibrotic foci. Functional experiments demonstrated the activation of fibroblasts in fibrotic ECM, which was alleviated by GPNMB-neutralizing antibodies or macrophage deletion in the ECM of silica-instilled mice. Moreover, the Serpinb2 expression level was increased in fibroblasts in fibrotic ECM, and the expression of CD44 was increased in silica-instilled mice. In conclusion, macrophage-derived GPNMB is trapped by fibrotic ECM during transport and may activate fibroblasts via the CD44/Serpinb2 pathway, thus leading to the further development of fibrosis.

[1] Jiangsu Provincial Key Laboratory of Critical Care Medicine, Zhongda Hospital, Department of Physiology, School of Medicine, Southeast University, Nanjing, Jiangsu 210009, China. [2] Department of Biomedical Engineering, Nanjing University of Aeronautics and Astronautics, Nanjing, Jiangsu 211106, China. [3] Key Laboratory of Environmental Medicine Engineering, Ministry of Education, School of Public Health, Southeast University, Nanjing, Jiangsu 210009, China. [4] Jiangsu Key Laboratory of Molecular and Functional Imaging, Department of Radiology, Zhongda Hospital, Medical School of Southeast University, Nanjing 210009, China. [5] School of Medicine, Xizang Minzu University, Xianyang, Shanxi 712082, China. [6] These authors jointly supervised this work: Wei Jiang, Jie Chao. ✉email: weijiang@nuaa.edu.cn; chaojie@seu.edu.cn

Pulmonary fibrosis (PF) is a type of chronic lung disease that leads to a progressive and irreversible decline in lung function[1,2]. The incidence of PF is increasing year by year, and the prognosis of PF is poor. A variety of lung diseases eventually lead to PF, and the mechanism of PF has not been fully elucidated. PF is characterized by chronic inflammation, excessive deposition of extracellular matrix (ECM) components, and lung tissue scarring resulting from aberrant wound healing in susceptible individuals[3–5]. Fibroproliferation is integral to host defense, and it ceases or regresses after mitigation or termination of the harmful stimulus[6]. However, a large amount of data have proven that PF progresses once it is established, suggesting that the initiation and sustained development of fibrosis can be independent of each other[7].

The ECM, which is present in all tissues, is a highly dynamic noncellular structure that serves as a physical scaffold for tissues and organs[8]. The lung ECM plays important roles in lung health from the earliest stages of development and throughout adulthood and is involved in the regulation of developmental organogenesis, homeostasis, and injury repair responses. In addition, the ECM is essential for cell functions such as survival, proliferation, migration, and differentiation via biochemical or biomechanical cues[9]. The ECM is composed of hundreds of proteins, including fiber-forming proteins such as collagens, laminins, fibronectin, elastin, glycoproteins, proteoglycans (PGs), and glycosaminoglycans, which form a complex three-dimensional matrix network. All cell types, especially fibroblasts, can synthesize and secrete matrix macromolecules, such as collagens and fibronectin, to participate in the construction of the ECM under the control of multiple signals[10,11]. Data show that components of the ECM can serve as ligands for cell receptors to interact with cells and then transmit signals that orchestrate cell behaviors[12]. Studies have found that ECM components interact with cells via their surface receptors, such as integrins, CD44, and cell surface PGs. The ECM can also bind and locally release growth factors and cytokines; for example, fibroblast growth factor (FGF) binds avidly to heparin and heparin sulfate, a component of many ECM PGs, and can be released from the ECM by protein degradation[9,13]. Therefore, the ECM serves as a localized reservoir for soluble growth factors, and many growth factors may utilize heparan sulfate as a cofactor when they bind to their signal receptors. The ECM plays key roles in cell fates.

A large amount of data have proven that aberrant proliferation of activated fibroblasts and abnormally excessive collagen deposition and ECM remodeling occur during PF progression. Once activated, fibroblasts could synthesize and secrete excessive matrix proteins, resulting in abnormal remodeling of the ECM[14–16]. Recently, it was reported that in idiopathic pulmonary fibrosis (IPF), a common type of clinical PF, abnormal fibrotic ECM (that is, the ECM after lung fibrosis is established) can promote the activation of normal fibroblasts through a positive feedback loop by downregulating the expression of antifibrotic miR-29 family members in normal fibroblasts and further aggravate PF[7], which may explain why fibrosis can be self-sustaining once it is established. However, the possible mechanism by which fibrotic ECM affects cell fate and leads to exacerbation of PF has not been well defined. To explore the contribution of fibrotic ECM to pathological gene expression in normal fibroblasts, we established a mouse model of chronic PF via intratracheal instillation of silica. Unlike the bleomycin-induced lung injury model, which exhibits transient physiological fibrosis[17], the silica-induced model presents progressive irreversible PF, allowing us to explore the true mechanism of the disease.

In summary, we observed changes in ECM in mice 56 days after silica instillation and analyzed the changes in the protein levels of ECM components by proteomics. Single-cell RNA sequencing (scRNA-seq) and spatial transcriptomics were used to analyze the possible contributors to these changes. Normal fibroblasts were transplanted into decellularized mouse lung ECM, and the effects of normal ECM and fibrotic ECM on cell behaviors and the possible mechanism were evaluated, in which macrophage-derived GPNMB is trapped by fibrotic ECM during transport and may activate fibroblasts via the CD44/Serpinb2 pathway, thus leading to the further development of fibrosis.

## Results

**Fibrotic ECM affects normal lung fibroblast function in silicosis mice.** To explore the contribution of the fibrotic ECM to normal fibroblast function, chronic PF was first established in mice via intratracheal instillation of silica. The CT imaging of the mouse chest and hematoxylin and eosin (H&E) staining showed obvious collagen deposition and PF in mice 56 days after silica instillation (Fig. 1a), which is consistent with previous reports[18,19]. Furthermore, a pulmonary function test was performed, and it was found that several pulmonary function indicators, including inspiratory capacity (IC, which is the volume inspired during slow inspiration), the expiratory reserve volume (ERV), forced vital capacity (FVC, which is the volume expired during fast expiration), and functional residual capacity (FRC), were decreased in the mice (Fig. 1b). These data showed that PF was well-established 56 days after silica instillation, suggesting that the model of progressive PF was successfully constructed and could be used for further research.

To better study the changes in lung ECM after PF and to explore the influences of lung ECM on fibroblast behaviors in vitro, mouse lung ECM was decellularized via treatment with detergents, salts, and DNase following the steps shown in Fig. 1c to avoid the effects of endogenous cellular components on the results. We first confirmed that all cellular and nuclear material was eliminated via the decellularization process. H&E staining, Masson trichrome staining, and DAPI staining showed that decellularized ECM reserved a similar architecture, albeit without the evidence of cells (Supplementary Fig. 1a–c). Western blotting and qPCR data showed that there were hardly any cellular proteins or mRNAs in the acellular ECM (Supplementary Fig. 1d–f). Subsequently, we cultured fibroblasts in the decellularized ECM, and the data showed that the cells survived (Supplementary Fig. 1). These results indicated that cells and nuclear material were cleared effectively by decellularization and that cells repopulated in the decellularized ECM, suggesting that the in vitro model could be used for our subsequent experiments.

We also investigated what happened to the lung ECM 56 days after silica instillation. H&E staining and Masson trichrome staining showed excessive collagen deposition and obvious structural changes in the fibrotic ECM (Fig. 1d). Moreover, COL1A1 (collagen I), FN1(fibronectin), COL3A1 (collagen III) and ACTA2/α-SMA exhibited obvious increases in expression, as well as abnormal deposition (Fig. 1d, Supplementary Fig. 2a, b). These results indicated that the lung ECM was altered after fibrosis was established. However, an analysis of Young's modulus showed that there was no obvious change in the stiffness of the ECM in NS- vs $SiO_2$-treated lungs (Fig. 1e). To explore whether fibrotic ECM influences normal fibroblast function, we assessed the viability, proliferation, migration, and activation of normal fibroblasts cultured in normal ECM and fibrotic ECM. The Cell Counting Kit-8 (CCK-8) assay was used to evaluate cell viability (proliferation and growth of cells) in different ECM, and the data showed that fibrotic ECM induced a clear increase in cell viability (Fig. 1f), as well as the fact that more cells were harvested from fibrotic ECM than from normal ECM (Fig. 1g). Fibroblasts can synthesize and secrete interstitial-associated proteins when activated[20], so we also studied mRNA expression levels in fibroblasts cultured in different types of ECM. The data showed that compared with normal ECM,

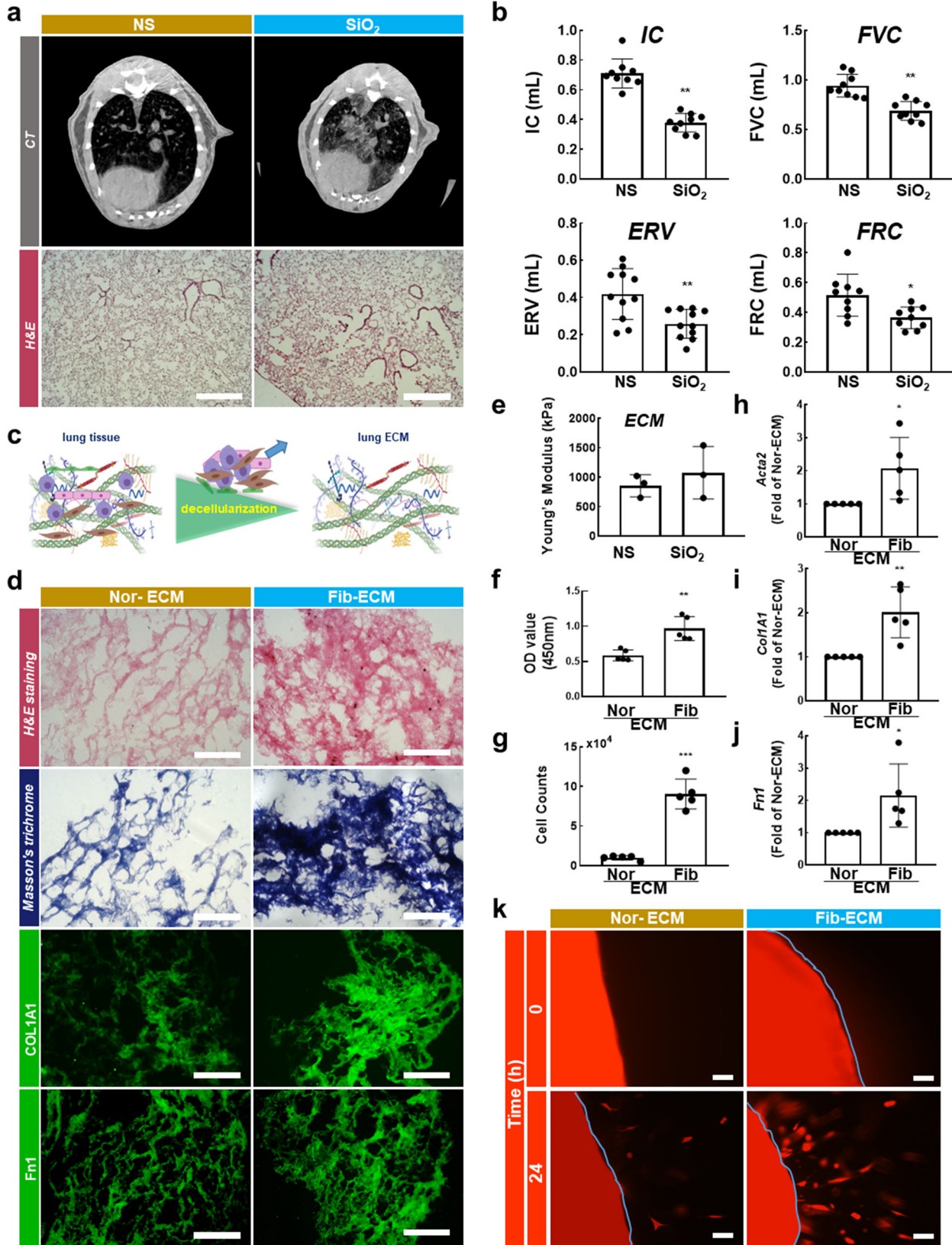

fibrotic ECM increased α-SMA, collagen I, and fibronectin mRNA expression levels in fibroblasts (Fig. 1h–j). In addition, fibrotic ECM promoted fibroblast migration (Fig. 1k, Supplementary Fig. 2c, d). All of the above mentioned experimental results indicated that the fibrotic ECM influences normal fibroblast function, thus leading to PF aggravation.

**Protein level in the fibrotic ECM changes obviously**. An understanding of the changes in the fibrotic ECM is necessary to determine the mechanism of fibroblast activation. The ECM is difficult to biochemically analyze because of its complexity and insolubility, and we strived to obtain detailed information about ECM components by using mass spectrometry-based proteomics.

**Fig. 1 Fibrotic ECM promoted normal lung fibroblast activation. a** CT images (top) and H&E staining (bottom) showing PF in mice after instillation of silica; scale bar = 200 μm. **b** The pulmonary function of mice in the two groups; $n \geq 10$ animals, **$p < 0.01$, *$p < 0.1$. **c** Schematic diagram of the protocol by which ECM was harvested. **d** H&E staining (top) and Masson's trichrome staining (bottom) showing that PF occurred in the ECM of mice after silica instillation; immunofluorescence staining showing that the expression of ECM components changed after silica instillation; scale bar = 200 μm. **e** Young's modulus of ECM in the two groups; $n \geq 3$ independent experiments. **f** The effect of fibrotic ECM on fibroblast viability; $n \geq 5$ independent experiments, ***$p < 0.001$. **g** The effect of fibrotic ECM on fibroblast proliferation; $n \geq 5$ independent experiments, ***$p < 0.001$. **h–j** The effect of fibrotic ECM on fibroblast activation; $n \geq 5$ independent experiments, **$p < 0.01$, *$p < 0.1$. **k** The effect of fibrotic ECM on fibroblast migration; $n \geq 5$ independent experiments, scale bar = 200 μm. NS, normal saline; Nor-ECM, normal ECM; Fib- ECM, fibrotic ECM.

An analysis of the quality of the proteomics results showed that the samples were reliable (Supplementary Fig. 3a–g). A total of 143 proteins with upregulated expression and 127 proteins with downregulated expression were identified in fibrotic ECM compared with normal ECM (Fig. 2a, b). Moreover, glycoprotein nonmetastatic melanoma protein B (GPNMB) was identified as one of the most highly upregulated proteins (Supplementary Table 1) that might be related to the pulmonary fibrosis process in silica-treated mice according to protein interaction network analysis (Fig. 2c, Supplementary Fig. 3h, i). Kyoto Encyclopedia of Genes and Genomes (KEGG) pathway analysis revealed that the upregulated proteins were related to pathways such as lysosome, coronavirus disease-COVID-19, phagosome, complement and coagulation cascades, apoptosis and antigen processing and presentation (Fig. 2d), and Gene Ontology (GO) enrichment analysis revealed that the upregulated proteins were enriched to positive regulation of response to external stimulus, activation of immune response, lysosome, extracellular matrix, endopeptidase activity, and peptidase activator activity involved in apoptotic process and so on(Fig. 2e). GPNMB was related to regulation of lymphocyte proliferation and receptor-ligand activity (Fig. 2f). The Western blotting analysis and immunofluorescence staining confirmed that the GPNMB level was substantially upregulated in the fibrotic ECM (Fig. 2g, h). Moreover, a proteomics analysis showed that the increased GPNMB level in the fibrotic ECM may be related to fibroblast activation.

**GPNMB protein trapped by fibrotic ECM is involved in the effect of ECM on lung fibroblast function.** Experiments were designed to assess the effect of GPNMB on fibroblast function in vitro. The viability of fibroblasts, as assessed by the CCK-8 assay, increased after GPNMB treatment (Fig. 3a). We harvested many more fibroblasts from the GPNMB-treated group (Fig. 3b), and the number of Ki67-positive cells was increased after treatment with GPNMB (Fig. 3c, d). In addition, fibroblasts exposed to GPNMB migrated much more quickly than those exposed to vehicle (Fig. 3e, f). Moreover, α-SMA, collagen I, and fibronectin mRNA expression levels were increased in the fibroblasts of the GPNMB-treated group (Fig. 3g). These data suggested that GPNMB can regulate normal fibroblast behaviors.

Subsequently, we asked whether the increased GPNMB level in the fibrotic ECM affects normal fibroblast behaviors. Normal fibroblasts were cultured in different types of ECM with a GPNMB-neutralizing antibody (Ab, 0.5 μg/mL), and then fibroblast migration was evaluated. The fibrotic ECM promoted fibroblast migration, and this effect was ameliorated by the GPNMB-neutralizing antibody (Fig. 3h, i). Therefore, an increased GPNMB level in the fibrotic ECM indeed influences normal fibroblast behaviors.

**GPNMB captured by fibrotic ECM may be derived from macrophages.** Although the increase in GPNMB level in the fibrotic ECM plays such an important role, it is unclear where this GPNMB is derived from. To identify the cellular origin of GPNMB, we obtained whole lungs from four groups of mice (the

NS-7 day group, SiO$_2$-7 day group, NS-56 day group, and SiO$_2$-56 day group) and performed scRNA-seq. The transcriptomic data for the four groups were normalized, and graph-based clustering was applied to analyze and compare the samples, which yielded 20 cell types (Fig. 4a). The cell type for each cluster was then annotated according to canonical cell markers from Cell-Marker (Supplementary Fig. 4a). The data showed that the relative percentage of macrophages was increased in the SiO$_2$-7 day group mice and in the SiO$_2$-56 day group mice (Fig. 4a, Supplementary Fig. 4b), thus implying that macrophages participated in both the inflammatory stage and in the fibrotic stage. GPNMB was mainly derived from macrophages according to the scRNA-seq analysis (Fig. 4b), and its expression was obviously increased in SiO$_2$-treated mice, including in SiO$_2$-56 day group mice (Fig. 4a, Supplementary Fig. 3j). Moreover, spatial transcriptomics showed that more macrophages accumulated in the focus area in SiO$_2$-7 day group mice and SiO$_2$-56 day group mice, where the expression of GPNMB upregulated by coincidence (Fig. 4c). In addition, immunofluorescence staining revealed that macrophage infiltration and localization of GPNMB in macrophages were increased in the lungs of SiO$_2$-56 day group mice compared with those of NS-56 day group mice (Fig. 4d). As a result, we speculated that GPNMB protein in fibrotic ECM, which showed an increase in protein levels, may be derived from macrophages.

**The effect of fibrotic ECM on lung fibroblast activation is alleviated by macrophage depletion.** To investigate the prospect that the cellular origin of GPNMB is macrophages, we administered clodronate liposomes (an agent for the depletion of macrophages) to mice every 7 days to remove macrophages because macrophages are constantly produced (shown in Fig. 5a). The data indicated that the administration of clodronate liposomes resulted in a marked decrease in the number of macrophages in mice (Supplementary Fig. 5a, b). The lungs were obviously damaged and became solidified after exposure to clodronate liposomes (Supplementary Fig. 5c). In addition, CT imaging of the chest showed that macrophage deletion by clodronate liposomes relieved PF induced by silica instillation (Fig. 5b). Moreover, H&E staining and Masson trichrome staining showed that silica instillation caused the obvious destruction of the alveolar structure and collagen deposition, which were mitigated by macrophage deletion (Fig. 5b, Supplementary Fig. 5d). H&E staining and Masson trichrome staining also revealed that macrophage deletion inhibited the obvious structural changes in fibrotic ECM and excessive collagen deposition (Fig. 5c).

To assess the effect of fibrotic ECM on normal fibroblast behaviors after macrophage deletion in mice, we cultured fibroblasts in ECM harvested from mice in the four groups. Cell viability and fibroblast number were decreased in the fibrotic ECM of silicosis mice subjected to macrophage deletion (Fig. 5d, e). Macrophage deletion also relieved fibroblast migration induced by fibrotic ECM (Fig. 5f, g). These data indicated that the effect of fibrotic ECM on lung fibroblast function is relieved by macrophage depletion.

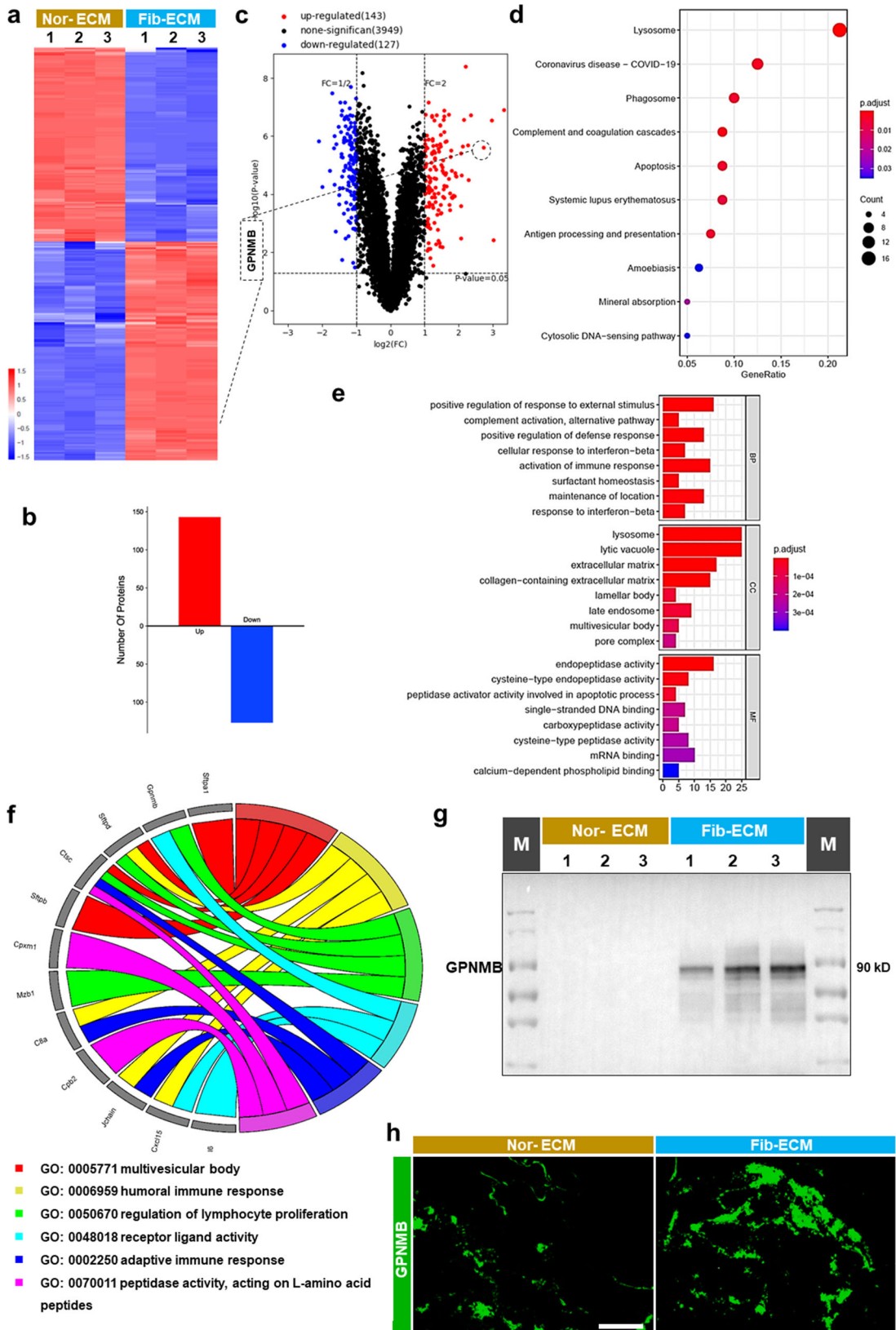

**Fig. 2 GPNMB protein level increased obviously in fibrotic ECM. a–c** Proteomics analysis showed obvious changes in the expression of ECM components with |log2(fc)| > 1 ($p < 0.05$) in mice from the two groups ($n = 3$ animals). **d–f** KEGG pathway analysis and GO enrichment analysis of the identified upregulated proteins. **g** Representative Western blot showing that the GPNMB protein level was increased in the ECM of silicosis mice; $n = 5$ independent experiments. **h** Immunofluorescence staining was used to assess the protein levels of GPNMB in fibrotic ECM; scale bar = 200 μm. Nor-ECM, normal ECM; Fib- ECM, fibrotic ECM.

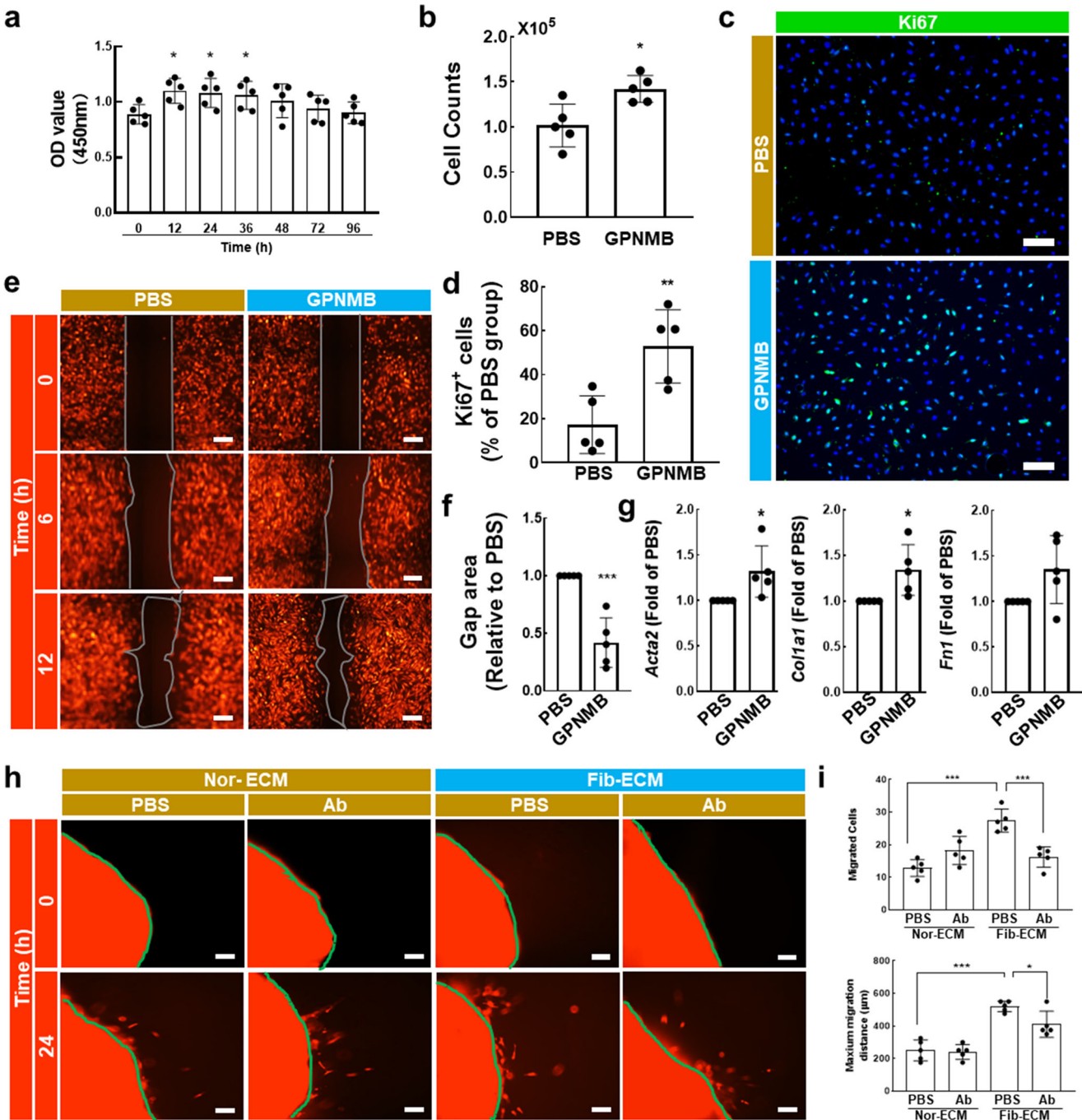

**Fig. 3 Increased GPNMB protein level was involved in the effect of fibrotic ECM on lung fibroblast activation. a** The effect of the GPNMB protein (0.1 μg/mL) on normal fibroblast viability; $n \geq 5$ independent experiments, ***$p < 0.001$, **$p < 0.01$. **b** The effect of GPNMB (0.1 μg/mL) on fibroblast proliferation; $n \geq 5$ independent experiments, *$p < 0.05$. **c, d** Immunofluorescence staining was used to evaluate the effect of GPNMB (0.1 μg/mL) on fibroblast proliferation; scale bar = 200 μm, $n \geq 5$ independent experiments, **$p < 0.01$. **e, f** The effect of GPNMB (0.1 μg/mL) on fibroblast migration; scale bar = 100 μm, $n \geq 5$ independent experiments, ***$p < 0.001$. **g** The effect of GPNMB (0.1 μg/mL) on fibroblast activation; $n \geq 5$ independent experiments, *$p < 0.05$. **h, i** The effect of fibrotic ECM and a GPNMB-neutralizing antibody (0.5 μg/mL) on fibroblast migration; scale bar = 100 μm, $n \geq 5$ independent experiments, ***$p < 0.001$, *$p < 0.05$. Nor-ECM, normal ECM; Fib-ECM, fibrotic ECM; Ab, GPNMB-neutralizing antibody.

**GPNMB protein level is decreased in fibrotic ECM after macrophage depletion**. We next evaluated GPNMB protein levels in fibrotic ECM after macrophage depletion to confirm the cellular origin of GPNMB in fibrotic ECM. As shown in Fig. 6a, a large number of macrophages infiltrated and colocalized with GPNMB in the lungs of mice 7 days and 56 days after silica instillation, and macrophage depletion decreased the GPNMB level, thus suggesting that macrophages increased in quantity and synthesized

and secreted GPNMB during the progression of PF. Concomitantly, macrophage depletion decreased GPNMB protein levels in the lung ECM of mice subjected to silica instillation, thus further confirming the macrophage origin of GPNMB (Fig. 6a, Supplementary Fig. 5e, f).

To further confirm our results, we treated RAW264.7 and THP-1 macrophages with silica and then observed the change in GPNMB protein levels. Western blot analysis showed that

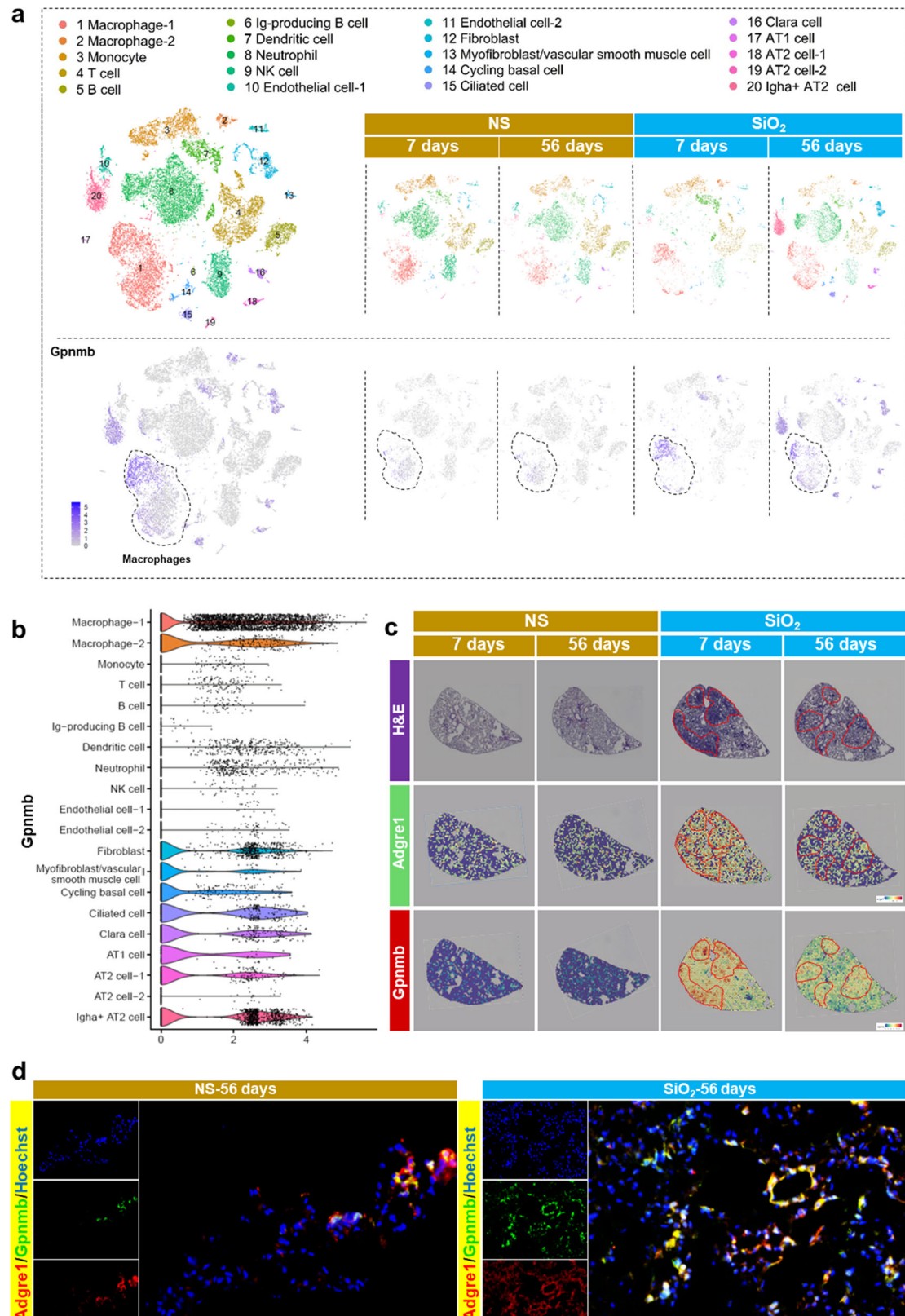

**Fig. 4 GPNMB in fibrotic ECM may have been derived from macrophages. a** Single-cell clustering and definitions. The cell type for each cluster was annotated according to canonical cell markers in CellMarker. Gpnmb expression in macrophages was increased in SiO$_2$-7 day group mice and SiO$_2$-56 day group mice. **b** Expression level of Gpnmbin various cell types. **c** The expression level of Adgre1 (cell marker of macrophages) and Gpnmb genes in different groups on the spatial plots. **d** Immunofluorescence staining of macrophages and GPNMB in NS-56 day group mice and SiO$_2$-56 day group mice; scale bar = 200 μm. NS, normal saline.

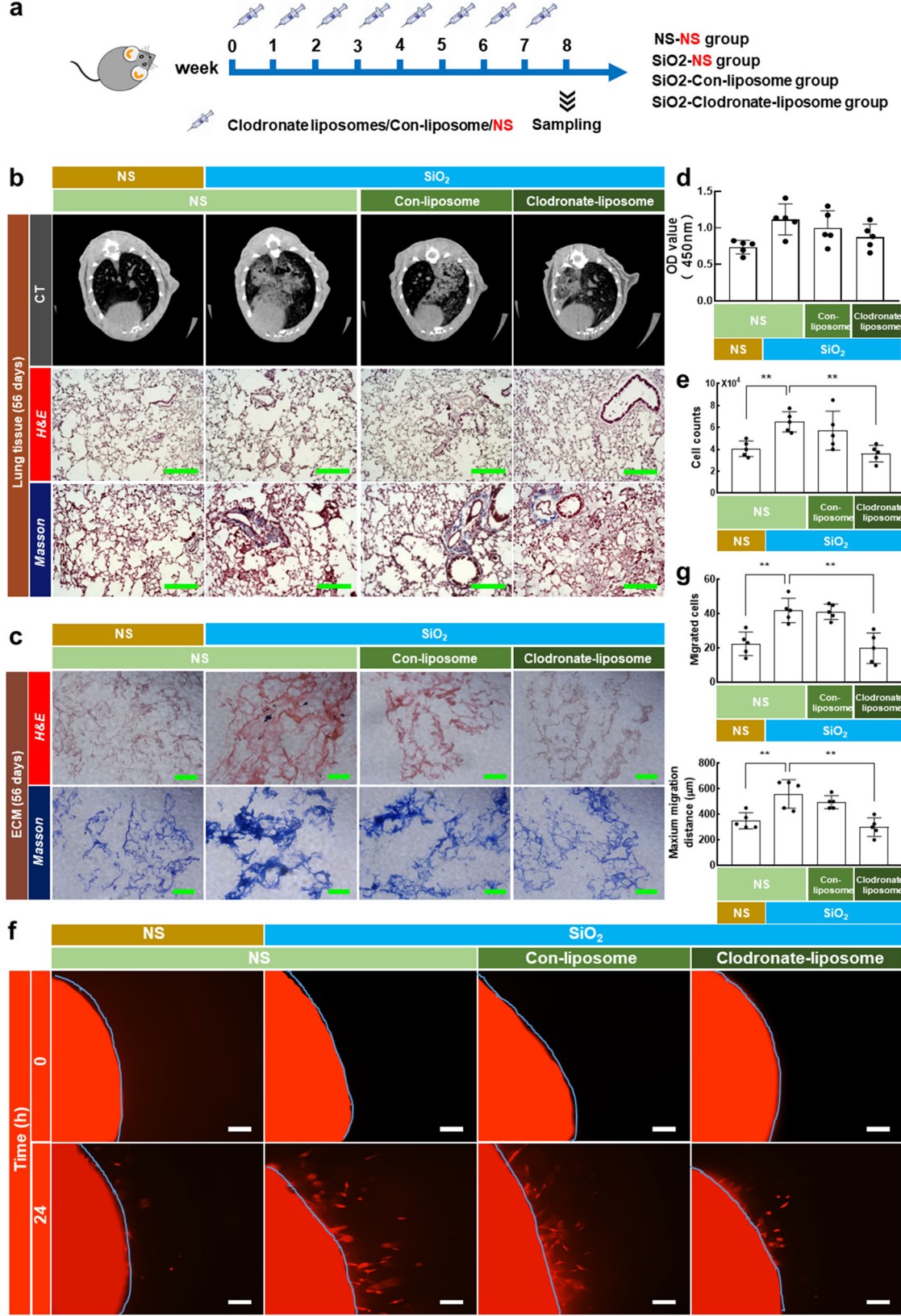

**Fig. 5 Macrophage deletion via clodronate liposomes relieved PF and fibrotic ECM affected normal fibroblast behaviors in silicosis mice. a** Design of the macrophage deletion experiment. **b** CT images, H&E staining, and Masson's trichrome staining of lung tissue; scale bar = 200 μm. **c** H&E staining and Masson trichrome staining of ECM; scale bar = 200 μm. **d, e** Changes in cell viability and fibroblast numbers in the fibrotic ECM of silicosis mice after macrophage deletion; $n \geq 5$ independent experiments, \*\*\*$p < 0.001$, \*\*$p < 0.01$. **f, g** The effect of macrophage deletion on fibroblast migration induced by fibrotic ECM; scale bar = 200 μm, $n \geq 5$ independent experiments, \*\*$p < 0.01$. NS, normal saline.

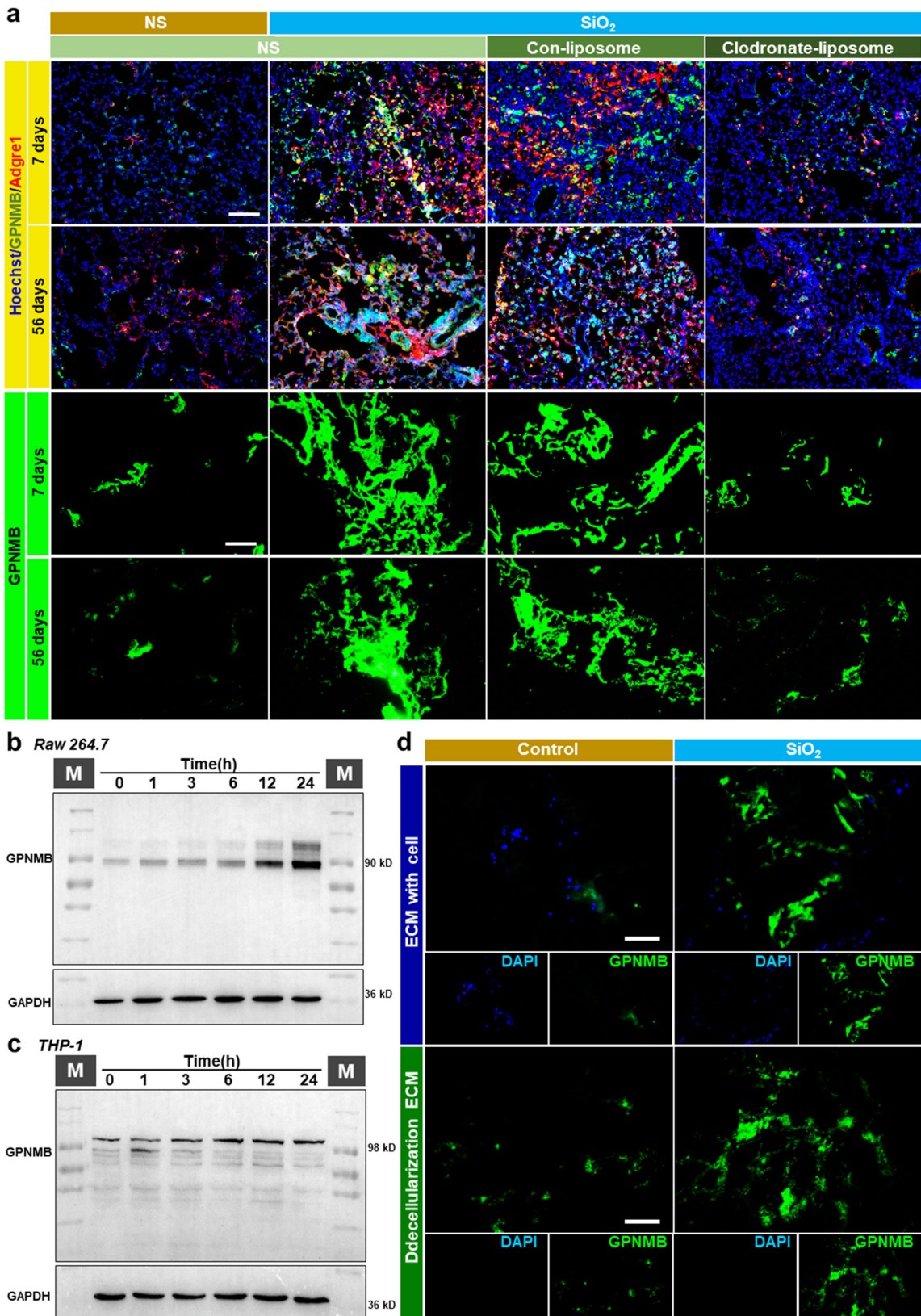

**Fig. 6 GPNMB protein level was decreased in fibrotic ECM after macrophage depletion. a** Immunofluorescence staining of macrophages and GPNMB in lung tissues from the four groups and GPNMB protein levels in ECM in the four groups; scale bar = 200 μm. **b, c** GPNMB protein levels in Raw 264.7 and THP-1 cells after silica treatment; $n \geq 5$ independent experiments. **d** Immunofluorescence staining of GPNMB in the ECM with or without Raw 264.7 cells; scale bar = 200 μm. NS, normal saline; Adgre1, macrophage markers.

GPNMB protein levels increased after silica treatment (Fig. 6b, c, Supplementary Fig. 6a, b). The macrophages isolated from murine also showed increased levels of GPNMB protein after silica treatment (Supplementary Fig. 6c, d).

However, whether GPNMB produced by macrophages following stimulation with silica could bind to the ECM was unclear. Thus, we cultured RAW264.7 cells in normal ECM in the presence of silica (Supplementary Fig. 7a). Surprisingly, we found that GPNMB levels in the ECM increased when RAW264.7 cells were cultured with silica (Fig. 6d); however, this result does not fully confirm that GPNMB produced by macrophages were indeed bound to the ECM, due to the fact that the effects of cellular components could not be excluded. Therefore, we decellularized ECM with which RAW264.7 cells had been cultured. Immunofluorescence staining showed that GPNMB produced by macrophages could reliably bind to the ECM (Fig. 6d), thus indicating that the GPNMB in the fibrotic ECM of silicosis mice was derived from macrophages.

**The CD44/Serpinb2 pathway is associated with changes in fibroblast function driven by fibrotic ECM**. The increase in GPNMB levels plays a nonnegligible role in the effect of the fibrotic ECM on fibroblast activation (as confirmed above); however, the related mechanism remains unclear. To explore the possible mechanism, we cultured normal fibroblasts in normal ECM and fibrotic ECM, and then gene expression in fibroblasts was assessed via transcriptome analysis. Sixteen genes with upregulated expression and 54 genes with downregulated expression were identified in fibroblasts cultured in fibrotic ECM compared with those cultured in normal ECM (Fig. 7a, b, Supplementary Table 2). The KEGG pathway analysis revealed that the upregulated genes enriched in pathways, such as rheumatoid arthritis, Th17 cell differentiation, vascular smooth muscle contraction, cell adhesion molecules, Rap 1 signaling pathway, cAMP signaling pathway (Fig. 7c); in addition, the GO enrichment analysis showed that mRNAs that showed upregulated expression in fibroblasts cultured in fibrotic ECM were enriched in various factors, including T cell activation, regulation of leukocyte cell–cell adhesion, protein complex involved in cell adhesion, G protein-coupled glutamate receptor binding and protein kinase C activity. (Fig. 7d). The protein interaction network analysis showed that Serpinb2, one of the most highly upregulated mRNA, could be regulated via CD44 (Fig. 7e), which is a receptor that mediates cell–cell and cell–matrix interactions through its affinity for HA and other ligands, such as collagens and matrix metalloproteinases (MMPs). Serpinb2, an inhibitor of extracellular urokinase plasminogen activator (uPA), was possibly related to fibroblast activation through CD44 Western blot analysis confirmed that Serpinb2 levels were substantially increased in fibroblasts treated with GPNMB (Fig. 7f, Supplementary Fig. 6e). The data (Fig. 7g) showed that the CD44 level was obviously increased in SiO$_2$-treated mice, especially in SiO$_2$-56 day group mice. Moreover, spatial transcriptomics revealed that CD44 levels were increased in the focus area in SiO$_2$-7 day group mice and SiO$_2$-56 day group mice (Fig. 7h). To further prove our hypothesis, fibroblasts were treated with GPNMB protein. To our surprise, Western blotting suggested that the CD44 protein level did not increase after GPNMB treatment (Supplementary Fig. 6f, g). Subsequently, fibroblasts were incubated with transforming growth factor-β (TGF-β), which is a significant inflammatory factor that promotes fibrosis formation, and Western blotting showed that CD44 expression was markedly increased (Fig. 7i, Supplementary Fig. 6h), thus indicating that increased CD44 expression may be a result of the inflammatory environment in PF. The siRNA-induced CD44

knockdown in fibroblasts suppressed cell migration induced by GPNMB treatment (Supplementary Fig. 8a). Furthermore, the levels of CD44 and Serpinb2 increased in the lungs of SiO$_2$-treated mice (Supplementary Fig. 9a, b). These results suggested that CD44/Serpinb2 is related to changes in fibroblast function driven by increased GPNMB levels in the fibrotic ECM.

When considering the importance of GPNMB, we analyzed the protein levels of GPNMB in patients with PF (data from the Gene Expression Omnibus database). The data showed that GPNMB levels were increased in PF patients (Supplementary Fig. 10a–c), thus further suggesting that GPNMB protein levels are related to progressive PF.

## Discussion

Despite the profound changes that the ECM undergoes during PF, there have been insufficient studies on the changes of proteins in the ECM during progressive PF, and the mechanism by which the fibrotic ECM affects cell fates, as well as the subsequent exacerbation of PF, has not been well defined. Herein, we found that an increase in the protein level of GPNMB (which was mainly derived from macrophages) in the fibrotic ECM promoted normal fibroblast activation via the CD44/Serpinb2 pathway, thus possibly accounting for the exacerbation of PF after its establishment.

Here, we used silica instead of bleomycin, which is commonly used to induce PF formation, to establish a progressive PF model in mice. Bleomycin can induce transient physiological fibrosis that gradually resolves within four to eight weeks[17,21]. Considering the purpose of our exploration, silica was used to construct a PF model for further study. Our data showed that the pulmonary function of mice was decreased 56 days after silica instillation, and CT imaging of the mouse chest and H&E staining showed extensive alveolar structure collapse and excessive collagen deposition, indicating well-established PF. Previous reports have also pointed out that PF develops 56 days after silica intratracheal instillation[22–24]. Therefore, we collected data and samples 56 days after silica treatment in our study.

As an important component of tissues and organs, the ECM can not only provide physical support for cells but also regulate cell behaviors. A large amount of data has proven that excessive collagen deposition and obvious ECM remodeling occur after PF and that these processes affect the cell functions, such as activation, proliferation, migration, and adhesion[6,25]. Fibrotic ECM downregulates the expression of antifibrotic miR-29 family members and activates a profibrotic positive feedback loop, leading to exacerbation of PF[6,7]. Therefore, considering the contribution of the ECM to cell behaviors, the initial damage might lead to the development of only a small fibrotic region in the lung and fibrosis of the ECM, which impacts and activates nearby fibroblasts, leading to surrounding lung tissue reconstruction and the spread of fibrosis. To explore the possibility of this phenomenon, we harvested pulmonary ECM from normal mice and silica-treated mice via decellularization and then cultured fibroblasts with the ECM. The structure of the harvested ECM and the successful removal of the cellular components were experimentally verified. The ECM we obtained retained a relatively complete and orderly structure, and intracellular components were undetectable, indicating that the ECM could be used for further study. In fact, the data indeed showed changes in fibrotic ECM, and the proliferation, migration, and activation of fibroblasts cultured in the silica-induced fibrotic ECM changed obviously. This aroused our curiosity. What alterations occurred in the ECM that led to changes in cell function?

Since the structure of the ECM is complex and most ECM components are insoluble macromolecular proteins, it is difficult

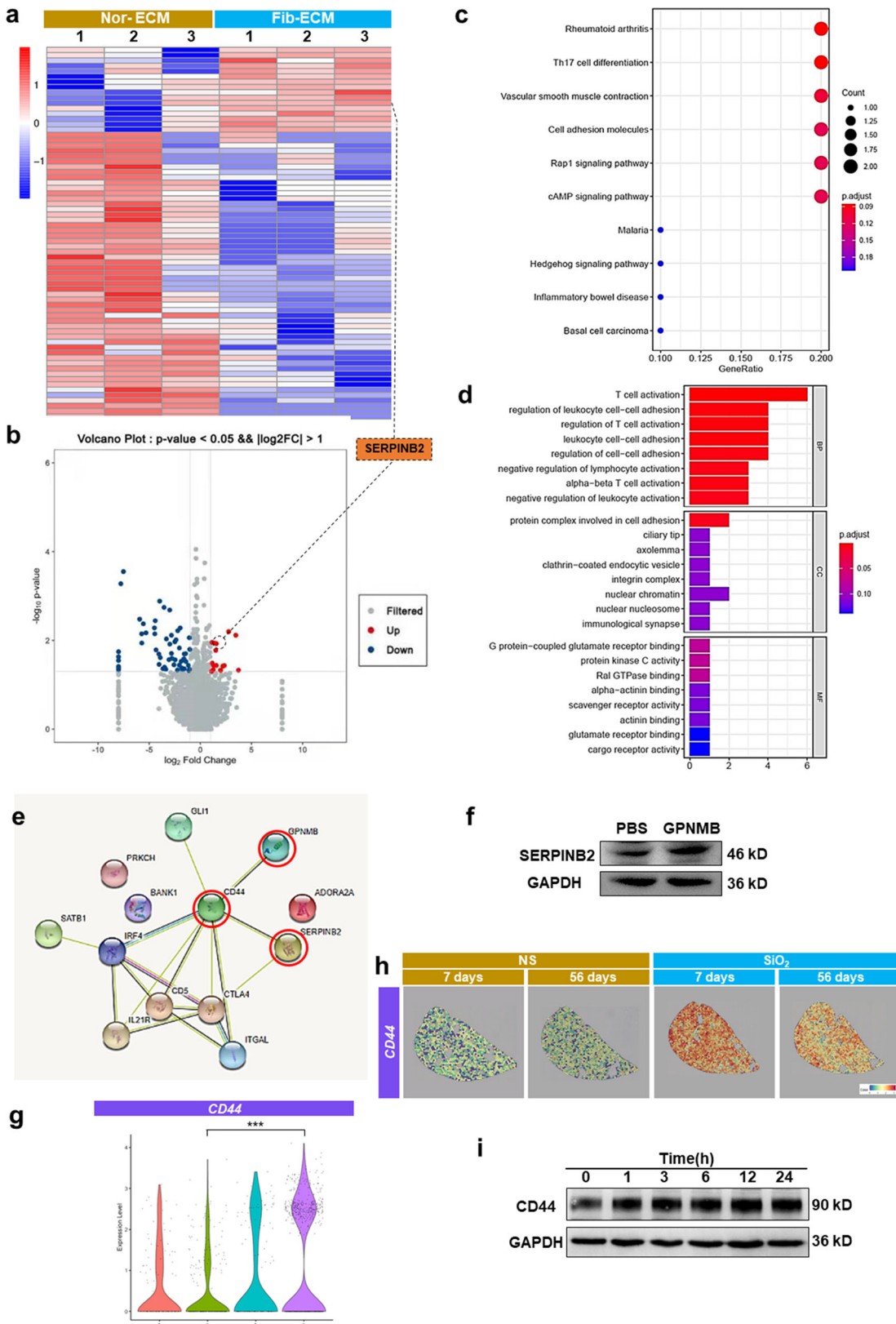

**Fig. 7 The CD44/Serpinb2 pathway is related to changes in fibroblast function driven by fibrotic ECM. a** Heatmap of differentially expressed mRNAs between fibroblast in different types of ECM. **b** Volcano plot of differentially expressed proteins between fibroblasts in different types of ECM; |log2(fc)| > 1, p < 0.05. **c** The KEGG pathways analysis of the upregulated mRNAs. **d** The GO enrichment analysis of the upregulated mRNAs. **e** Protein interaction network analysis between GPNMB and Serpinb2. **f** The change in Serpinb2 protein levels after GPNMB treatment; n ≥ 5 independent experiments. **g** The expression level of CD44 gene in different groups on the spatial plots. **h** The change in CD44 levels in the four groups, as determined by spatial transcriptomics. **i** CD44 protein levels in fibroblasts after TGFβ1 (5 ng/mL) treatment; n ≥ 5 independent experiments. NS, normal saline; Nor-ECM, normal ECM; Fib- ECM, fibrotic ECM.

to explore this structure. We used mass spectrometry-based proteomics to assess and analyze the changes in protein expression in lung ECM from normal mice and silica-treated mice. Fortunately, we were able to map the changes in ECM protein expression between the two groups. A total of 143 proteins with upregulated expression and 127 proteins with downregulated expression were identified in fibrotic ECM compared with normal ECM. Through protein interaction network analysis, we confirmed that GPNMB trapped by fibrotic ECM might play a nonnegligible role in the activation of fibroblasts. GPNMB, a transmembrane glycoprotein that exists in a soluble form, is widely expressed in many cell types and participates in the regulation of multiple functions in various tissues. GPNMB is also known as osteoactivin because of its role in osteoblast differentiation and increasing bone mineral deposition[26]. GPNMB indeed affected cell proliferation, migration, and activation, but these findings were not enough to confirm that GPNMB trapped by fibrotic ECM contributed to these changes. To explore the role of GPNMB captured by fibrotic ECM, we used neutralizing antibodies to assess the effect of GPNMB on the fibrotic ECM. The effect of the fibrotic ECM on normal fibroblasts was alleviated after treatment with a GPNMB-neutralizing antibody, confirming the contribution of GPNMB trapped in the fibrotic ECM to these changes.

Due to the fact that the increase in the GPNMB level plays an important role in the effect of fibrotic ECM, there is a question as to where the excess GPNMB arises from. By combining mass spectrometry-based proteomics, scRNA-seq, and spatial transcriptomics, we confirmed that GPNMB might be derived from macrophages. Macrophages are activated in response to silica, leading to changes in cell morphology and cell secretion patterns, which play roles in the progression of PF[27–29]. We employed clodronate liposomes to remove macrophages in mice[30,31], and H&E staining and Masson trichrome staining showed that PF was alleviated after clodronate liposome administration. Moreover, the ECM from silica-treated mice subjected to macrophage deletion had a less marked effect on the viability, proliferation, and migration of normal fibroblasts and exhibited a decrease in GPNMB levels. These data suggested that GPNMB in the fibrotic ECM might be derived from macrophages and involved in regulating cell behaviors. Furthermore, scRNA-seq also showed increased GPNMB mRNA levels in other cells after silica instillation, such as AT-2 cells, and we mainly focused on macrophage-derived GPNMB according to our included data. However, our study could not exclude the fact that GPNMB from other cells may also be involved in the process of pulmonary fibrosis induced by silica.

ECM components can serve as ligands to allow cell receptors to interact with cells and then transmit signals that orchestrate cell behaviors[32,33]. How does GPNMB in fibrotic ECM regulate cell function? Transcriptome analysis showed that Serpinb2 mRNA expression was upregulated in fibroblasts in fibrotic ECM compared with those in normal ECM. Serpinb2, a regulator of inflammatory processes, is involved in aging, injury, and repair[34]. Studies have shown that the activity of uPA is associated with lung fibrosis and that uPA overexpression enhances the resolution of established lung fibrosis[35]. Serpinb2 is an inhibitor of extracellular uPA and might be related to lung fibrosis development. Our data showed that Serpinb2 levels were increased in GPNMB-treated fibroblasts. Studies have found that ECM components interact with cells via their surface receptors, and in both forms, GPNMB has been shown to interact with many different partners, such as integrins, heparan sulfate PG, tyrosine kinase receptors, and transporters. CD44 is a transmembrane glycoprotein that is widely expressed on various cell types and is a receptor for GPNMB[26,36,37]. CD44 plays an important role in various cellular events, including activation, proliferation, migration, and adhesion, via cell–matrix or cell–cell interactions. scRNA-seq and spatial transcriptomics showed that CD44 expression was increased in the lungs of silica-treated mice. However, the CD44 level did not increase after exposure to GPNMB. All kinds of cells take part in tissue repair and reconstruction. They play diverse roles, including releasing many inflammatory factors, such as TGF-β1, to form an inflammatory environment at the site of lung injury[35]. TGF-β1 is an important fibrogenic factor, and its expression has been found to increase during the fibrosis process[38]. We wondered whether the inflammatory environment during fibrosis leads to upregulation of CD44 expression. Therefore, we treated fibroblasts with TGF-β1, and the CD44 expression level indeed increased, consistent with our hypothesis. Our data showed that fibrotic ECM captured GPNMB derived from macrophages, leading to a local increase in the protein level of GPNMB and that some inflammatory factors released by cells during fibrosis increased CD44 levels in normal fibroblasts around the fibrotic ECM. Therefore, GPNMB trapped by fibrotic ECM encountered CD44 in normal fibroblasts and Serpinb2 levels increased, leading to cell activation and further promotion of PF development.

These data indicated that the additional GPNMB protein derived from macrophages in fibrotic ECM increased the Serpinb2 level in normal fibroblasts via CD44, leading to changes in cell behaviors that promoted the progression of PF. Drugs that can inhibit the abnormal increase in GPNMB levels in the ECM may have the potential to treat PF.

## Methods

**Mice**. C57BL/6 mice (male, 20 ± 2 g) purchased from Hangzhou Ziyuan Experimental Animal Co., Ltd. (Hangzhou, China), were housed at 23 °C and 50% humidity on a 12-h light/dark cycle with free access to food and water. All animal procedures were approved by the Laboratory Animal Care and Use Committee of Southeast University (20210106011) and were performed in strict accordance with the National Institutes of Health Guide for the Care and Use of Laboratory Animals.

**Establishment of a mouse model of PF**. Silica (Sigma) was sterilized and suspended (50 mg/ml) in sterile normal saline (NS). The mice were intratracheally administered the silica suspension (100 μL) after anesthesia with pentobarbital sodium (1%, 50 mg/kg). Mice in the NS group were given an equivalent volume of NS in the same way.

**Macrophage depletion**. The mice were randomly divided into the following four groups: the NS (normal saline)-NS group, SiO₂-NS group, SiO₂-Con liposome group, and SiO₂-clodronate liposome group. The mice in the SiO₂-NS group, SiO₂-Con liposome group, and SiO₂-clodronate-liposome group were intratracheally administered 100 μL of silica suspension (50 mg/mL) after anesthesia with pentobarbital sodium (1%, 50 mg/kg). The mice in the NS-NS group were given an equivalent volume of NS in the same way. The mice in the SiO₂-clodronate liposome group were administered clodronate liposomes every 7 days via tail vein injection to induce macrophage depletion. The mice in the SiO₂-Con liposome group were given an equivalent volume of control liposomes in the same way. For comparison, the mice in both the NS-NS group and SiO₂-NS group were given an equivalent volume of NS in the same manner.

**Decellularization of lung matrices**. Decellularized ECM was harvested based on a previous report with some modifications[39]. Pieces of lung tissue were frozen at −70 °C, and 200 μm slices of frozen tissue were made with a freezing microtome. The tissue slices were successively decellularized in lysis buffer (1% SDS in ddH₂O), 1% Triton X-100 (diluted with ddH₂O), and NaCl (1 M) at room temperature. Then, the tissue slices were incubated in solution containing DNase (20 μg/ml) plus MgCl (4.2 mM) at 37 °C for 1 hour. The decellularization process was terminated after aspiration of DNase.

**Sample preparation for proteomic analysis of the ECM (PXD028194)**
*ECM protein enrichment and LC–MS analysis*. ECM proteins cannot be purified because of their insolubility and are considered the remaining proteins after cytosolic, nuclear, membrane, and cytoskeletal proteins are removed. ECM protein enrichment was performed with a Cytosol/Nucleus/Membrane/CytoSkeleton (CNMCS) Compartmental Protein Extraction Kit (Millipore, Temecula, CA, USA).

ECM protein was extracted from NS-56 day group (n = 3, named con111, con116, and con117) and SiO₂-56 day group (n = 3, named M80, M101, and M107) tissues, and all extraction processes were carried out in accordance with the instructions. Equal amounts of each labeled sample were mixed, and an appropriate amount of protein was used for chromatographic separation. All the samples were analyzed by LC–MS.

*Analysis of LC–MS/MS data.* The raw LC–MS/MS data were processed using Proteome Discover 2.4 (Thermo, USA). According to the unique peptide ≥1, any group of samples with a protein expression value ≥50% was retained. Then, the missing values were imputed with the mean protein expression in the corresponding group. Next, the data were median normalized and log2-transformed to identify candidate proteins. Then, we performed statistically analyzed, visualized, and plotted these proteins using R software (version 4.2) ggplot2 package (version 3.2.2), including by principal component analysis (PCA), sample correlation analysis, sample hierarchical cluster analysis, visualization of the data after standardization and density plotting.

We analyzed the data for the candidate proteins by Student's t test to identify proteins that showed significant differences in expression between the NS-56 day group and SiO₂-56 day group. The fold change (FC) was used to evaluate the expression level of individual proteins between samples. The $p$-value calculated using the t test to determine the significance of the difference between samples. The screening conditions were FC ≥ 2.0 and $P ≤ 0.05$. A clustering heatmap constructed using the pheatmap package (version 1.0.12) in R software (version 4.2) was used for quality control of the standardized experimental data and to visualize the differential expression data. Generally, samples from the same group appeared in the same cluster.

For the identified proteins, annotation information was extracted using the UniProt database. After identification of the differentially expressed proteins (| log2(FC)| > 1, $p < 0.05$), GO and KEGG functional enrichment analyses of upregulated proteins were performed with the ClusterProfiler package (version 3.16.1) in R software (version 4.2). (40-43). The mass spectrometry proteomics data have been deposited to the ProteomeXchange Consortium (http://proteomecentral. proteomexchange.org) via the iProX partner repository with the dataset identifier PXD028194.

### Spatial transcriptomics (GSE183683)
*Sample collection.* Mice with obvious fibrotic lesions on CT imaging were identified, and their lung tissues were trimmed near the hilum in the horizontal direction and frozen in OCT on dry ice as quickly as possible. These samples were stored at −80 °C before the next step.

*Staining and imaging.* The cryosections were sliced (a thickness of 10 μm) and placed on a Gex array, which was then placed in a Thermocycler Adaptor with the active surface facing up and incubated for 1 min at 37 °C. Then, the sections were fixed with methyl alcohol for 30 min at −20 °C and stained with H&E (Eosin, Dako CS701, Hematoxylin Dako S3309, bluing buffer CS702). Brightfield images were captured with a Leica DMI8 whole-slide scanner at 10x resolution.

*Permeabilization and reverse transcription.* Spatial gene expression was analyzed out using a Visium spatial gene expression slide and reagent kit (10x Genomics, PN-1000184). For each well, a slide cassette was used to create leakproof wells to allow the addition of reagents. The sections were incubated with 70 μL of permeabilization enzyme at 37 °C. For the NS-7 day, SiO2-7 day, and NS-56 day groups, the incubation time was 24 min, while the incubation time for the SiO2-56 day group to induce severe lung fibrosis. Each well was washed with SSC (100 μL), and RT master mix (75 μL) was added for cDNA synthesis.

*cDNA library preparation for sequencing.* After first-strand synthesis, the RT master mix in each well was replaced with KOH (0.08 M, 75 μL). After incubation at room temperature for 5 min, the slices were washed with EB buffer (100 μL), and then Second Strand Mix (75 μL) was added for second-strand synthesis. cDNA amplification was performed on a S1000TM Touch Thermal Cycler (Bio-Rad). Visium spatial libraries were constructed using the Visium spatial library construction kit (10x Genomics, PN-1000184) according to the manufacturer's instructions. The final libraries were sequenced using an Illumina NovaSeq6000 sequencer with a sequencing depth of at least 100,000 reads per spot using 150 bp (PE150) read strategy (performed by CapitalBio Technology, Beijing).

### Single-cell sequencing (GSE183682)
*Sample collection.* The inclusion criteria for the model group were the same as those used for spatial transcriptomics. Lung samples for scRNA-seq were collected from four groups of mice, namely, the NS-7 day, SiO₂-7 day, NS-56 day, and SiO2-56 day groups. The whole lungs of each mouse were removed within 2 min of euthanasia and quickly washed in precooled PBS 3 times.

Cell capture and cDNA synthesis: Whole lung tissues were cut into small pieces (approximately 1 mm) and dissociated into single cells using a Lung Dissociation Kit (Miltenyi Biotech, 130-095-927, Germany). With the Single-Cell 5' Library and

Gel Bead Kit (10x Genomics, 1000169) and Chromium Single-Cell G Chip Kit (10x Genomics, 1000120), cells suspensions (300–600 living cells per microliter determined by CountStar) were loaded onto a Chromium single-cell controller (10x Genomics) to generate single-cell gel beads in emulsion (GEMs) according to the manufacturer's protocol. In short, single cells were suspended in PBS containing 0.04% BSA. Approximately 20,000 cells were added to each channel, and the target cell recovery was estimated to be approximately 10,000 cells. Captured cells were lysed, and the released RNA was barcoded through reverse transcription in individual GEMs. Reverse transcription was performed on a S1000TM Touch Thermal Cycler (Bio-Rad) at 53 °C for 45 min followed by 85 °C for 5 min and a hold at 4 °C. cDNA was generated and then amplified, and quality was assessed using an Agilent 4200 system (performed by CapitalBio Technology, Beijing).

scRNA-seq library preparation: The scRNA-seq libraries were constructed using the Single-Cell 5' Library and Gel Bead Kit, Single Cell V(D)J Enrichment Kit and Human T Cell (1000005) and Single Cell V(D)J Enrichment Kit according to the manufacturers' instructions. The libraries were sequenced using an Illumina NovaSeq6000 sequencer with a sequencing depth of at least 100,000 reads per cell with a paired-end 150 bp (PE150) read strategy (performed by CapitalBio Technology, Beijing).

*Data preprocessing*
Analysis of scRNA-seq data: Cell barcode filtering, alignment of reads, and UMI counting were performed with Cell Ranger 4.0.0 (https://www.10xgenomics.com/). Further analysis was mainly conducted with the R package Seurat v3.2.2 based on the official tutorial. For quality control and filtering, we used R package DoubletFinder to effectively detect and remove doublets. The doublet rate (the nExp parameter in DoubletFinder) was estimated from the 10x Chromium users guide according to the number of recovered cells. Then, cells with detected genes less than 500 or with mitochondrial gene content >20% were excluded. All genes analyzed were present in greater than 10 cells. The scRNA-seq data was normalized with LogNormalize(scale factor 10,000). Two thousand highly variable genes were identified and data were scaled. We performed principal component analysis (PCA) for primary dimensionality reduction. The number of principal components was selected based on elbow plot and we chose 30 dimensions for downstream analysis. Batch effects among four samples were removed with Harmony. Thirty-three Clusters were identified by the FindNeighbors (based on KNN graphs) and FindClusters (based on Louvain method, resolution = 1) functions in Seurat. Specific marker genes were used to define the 33 clusters into 20 cell types. Harmony embeddings were used as input for t-Distributed Stochastic Neighbor Embedding (tSNE), which allows data visualization in a two-dimensional space. We calculated the differential expression levels of genes using the FindMarkers function based on the Wilcoxon rank sum test with default parameters.

Cell type annotation: Cell types were determined by clustering and marker gene expression.

### Cell culture and treatment.
Mouse lung fibroblast cells (MLg [Mlg 2908] [ATCC® CCL206™]) were cultured in DMEM containing fetal bovine serum (FBS, 10%) and penicillin–streptomycin (PS). After they reached 80–90% confluency, the cells were exposed to GPNMB protein (R&D Systems) or TGF-β1 (Kingsley Biotechnology) and then used for subsequent experiments. RAW264.7 (ScienCell) and THP-1 (ATCC) cells were cultured in DMEM containing 10% FBS and penicillin–streptomycin at 37 °C in a 5% CO₂ incubator. After they reached 80–90% confluency, the cells were exposed to silica (Sigma) and then used for subsequent experiments.

### Isolation and culture of murine primary macrophages.
Bone marrow-derived macrophages were isolated and cultured as previously described with some modifications. Mice were euthanized after which the femoral and tibial bones were opened to obtain Naive macrophages. Subsequently, the cells were cultured in RPMI 1640 (Corning, USA) containing 10% FBS with murine recombinant macrophage colony-stimulating factor (M-CSF) for 7 days. After 7 days, the cells were harvested for further experiments.

### Reseeding acellular matrices.
Fibroblasts were seeded dropwise on acellular slices and cultured in DMEM containing FBS (10%) and PS The medium was changed every other day. To observe the effect of GPNMB protein on ECM, a neutralizing antibody (R&D Systems) was used to reduce the activity of GPNMB on ECM.

### CT scanning.
CT imaging (Hiscan XM Micro CT, Suzhou Hiscan Information Technology Co., Ltd.) of the mouse chest was performed after anesthesia with inhaled isoflurane (induction: 3–44%, maintenance: 1–1.5%).

The X-ray tube settings were 60 kV and 133 μA, and images were acquired at 50 μm resolution. A 0.5° rotation step through a 360° angular range with 50 ms exposure per step was used. The images were reconstructed with Hiscan Reconstruct software (Version 3.0, Suzhou Hiscan Information Technology Co.,

Ltd.) and analyzed with Hiscan Analyzer software (Version 3.0, Suzhou Hiscan Information Technology Co., Ltd.).

**Pulmonary function test.** The mice with anesthetized with pentobarbital sodium, a tracheal catheter was inserted and fastened to the trachea, and then the trachea was exposed to assess pulmonary function. Then, IC, ERV, FVC, and FRC were tested with the Forced Manoeuvres System (EMMS, Hants, UK). Each mouse underwent three measurements before being sacrificed for collection of lung samples for further study.

**Western blotting.** RIPA buffer (Beyotime) containing a protease inhibitor cocktail (Roche) was used to isolate protein from samples. Primary antibodies against collagen I (1:1000, BioWorld), α-SMA (1:1000, Proteintech), FN1 (1:1000, Proteintech), GPNMB (1:1000, Abcam), Serpinb2 (1:1000, Affinity), and CD44 (1:1000, Proteintech) were used. GAPDH (1:1000, BioWorld) was used as an internal reference protein.

**Real-time quantitative RT–PCR (qRT–PCR).** qRT–PCR was performed to measure relative mRNA expression. Total RNA was isolated from samples using TRIzol reagent (Thermo Fisher Scientific). The concentrations of the samples were normalized, the RNA was reverse transcribed into cDNA, and then qRT–PCR was performed. The expression of target mRNAs was normalized to the mRNA expression of the recognized marker gene GAPDH.

**CCK-8 assay.** CCK-8 assay kit (Sigma-Aldrich) was employed to evaluate the viability of fibroblasts after stimulation. Cells were incubated with CCK-8 reagent at 37 °C for 1-4 h, and then the viability of the cells was measured with a microplate reader at 450 nm.

**Cell migration assay.** A 2D scratch assay was performed to assess cell migration capacity after treatment. Images of the scratch were captured at 0 h, 6 h, and 12 h after addition of fresh medium. A 3-dimensional (3D) migration assay was used to evaluate cell migration. Briefly, a standard fibroblast-populated, 3D collagen matrix (FPCM) was incubated in the attached state for 48 h, and then the FPCM was removed from the culture dish and embedded in a fresh acellular collagen matrix. Before 1 ml of DMEM containing 10% FBS was added to the well, the matrix system was allowed to polymerize for 1 h at 37 °C in a 5% $CO_2$ incubator. Cell migration from the nested FPCM and into the cell-free matrix was imaged with a fluorescence microscope at different times. To quantify cell migration, the number and the maximum migration distance of cells that migrated out from the nested matrix to the cell-free matrix were observed[40].

**Histopathological analysis.** Frozen slices were subjected to H&E (Biyun Tian) and Masson's trichrome (Biyun Tian) staining according to the manufacturer's instructions.

**Immunofluorescence staining.** Frozen lung sections or cells were incubated with primary antibodies against collagen I (1:200, BioWorld), α-SMA (1:200, Proteintech), FN1 (1:200, Proteintech), collagen III (1:200, BioWorld), GPNMB (1:300, Abcam), or Adgre1 (1:300, Abcam). Cy3-conjugated (1:500, Invitrogen) and/or Alexa Fluor 488-conjugated (1:300, Invitrogen) secondary antibodies were added to the sections or cells. The cell nuclei were labeled with Hoechst 33342 or DAPI if needed.

**Statistics and reproducibility.** All of the data are presented as the mean ± standard error of the mean (SEM), with sample sizes and numbers of repeats indicated in the figure legends. Comparisons between two groups were analyzed using a two-tailed Student's t test. One-way ANOVA with the multiple comparisons Bonferroni test was used to compare multiple groups. Differences were considered significant at $p < 0.05$. All analyses were conducted using GraphPad Prism version 8.0. The sample sizes, number of animals or replicates, and statistical comparison groups are indicated in the Figure legends.

**Reporting summary.** Further information on research design is available in the Nature Portfolio Reporting Summary linked to this article.

## Data availability

The mass spectrometry proteomics data have been deposited to the ProteomeXchange Consortium (http://proteomecentral.proteomexchange.org) via the iProX partner repository with the dataset identifier PXD028194. The single-cell sequencing of mouse lung tissue data have been deposited to the GEO datasets (https://www.ncbi.nlm.nih.gov/geo/) with the dataset identifier GSE183682. The Spatial transcriptomics of mouse lung tissue data have been deposited to the GEO datasets (https://www.ncbi.nlm.nih.gov/geo/) with the dataset identifier GSE183683. The uncropped and unedited blots to make graphs are included in the supplementary data file (Figs. 11–15 in Supplementary Data 1).

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

## Acknowledgements

This study is the result of work that was partially supported by resources and facilities at the Core Laboratory at the Medical School of Southeast University. This work was also supported by grants from the National Key R&D Program of China (2022YFC2504403), The National Natural Science Foundation of China (Nos. 81972987, 81773796, 81700068, 81803182 and 81901809), and the Jiangsu Provincial Key Laboratory of Critical Care Medicine (JSKLCCM-2022-02-005).

## Author contributions

J.W., X.Z., M.L., and M.Y. performed the experiments, interpreted the data, prepared the figures, and wrote the manuscript. J.W. wrote the revised manuscript. J.Y., M.L., W.L., S.W., and Y.C. performed the experiments and interpreted the data. W.J. designed the experiments, interpreted the data, and wrote the manuscript. J.C. provided the laboratory space and funding, designed the experiments, interpreted the data, wrote the manuscript, and directed the project. All of the authors have read, discussed, and approved the final manuscript.

## Competing interests

The authors declare no competing interests.
