## [Peer Review File · Communications Biology]

Reviewers' comments:

Reviewer #1 (Remarks to the Author):

Wang et al investigate the contribution of fibrotic ECM on fibroblast gene expression and phenotype in context of an experimental model of fibrosis. A major theme of the study is the development of an interesting model system to address how ECM associated with pulmonary fibrosis impacts cell fate. Pulmonary fibrosis is a significant unmet need worldwide. Improved model systems to study disease pathogenesis and the biology of cells associated pulmonary fibrosis are needed. Here the authors instigate pulmonary fibrosis by intratracheal instillation of silica into mice. Lung fibrosis is evaluated at day 7 and day 56 post silica administration. The authors demonstrate through lung functional assays that silica administration does indeed reduce lung function in the mice. Lung fibrotic tissue is used for downstream analysis. Interestingly lung tissue is also subjected to decellularization and subsequently seeded by mouse fibroblasts to investigate how the ECM of normal lung tissue compares to ECM of fibrotic lung tissue in terms of fibroblast gene/protein expression and phenotype. This is a strength of the study. The authors perform proteomic and transcriptomic assays leading to the identification of GPNMB as potentially involved in the fibrotic phenotype post silica administration. After cell based and expression assays the authors conclude that macrophage derived GPNMB promotes lung fibrosis. Overall the study is interesting, exploits useful approaches and has potential to provide impact. Concerns are mainly centered around interpretation of the data and inflation of the strength of the data to draw conclusions. Specific comments are below.

Major Comments:

1. A major concern is the tendency of the authors to over-interpret the data leading to some conclusions are not supported. Some examples are below
 - Fig 1E: the data do not indicate a change in stiffness in NS vs SiO₂ treated lungs yet the authors on line 156 indicate that stiffness is elevated in lungs from SiO₂ treated mice.
 - Fig 2C: the authors state on Line 180 that GPNMB is the most highly upregulated protein in PF. This is not true according to the data presented in Fig. 2C.
 - Fig 4A: the authors state in lines 223-224 that GPNMB was obviously increased in SiO₂-56 day 56 group mice, but this is not clear at all based on the data shown in Fig. 4A.
 - Fig 4: in general the authors discount the potential contribution of other cell types that express GPNMB.
 - Fig 7: the authors mine the omics data to craft a potential pathway that fits the change in GPNMB expression. There is no functional data to support that Serpinb2 or CD44 are functionally involved based on the data provided. The authors should soft pedal their conclusions. The data shown in Fig. 7B-K are modest at best.

None of these are fatal flaws. The data are useful and the authors, in my opinion, should downplay hard conclusions. They data are very nice but do not provide meaningful functional demonstration of the contribution of GPNMB or Serpinb2/CD44 to lung fibrosis. The data are descriptive and that is ok and useful for the field.

2. More detail is required on how the 3D migration assay is performed and quantified (Fig 1K, 3H, 5F).
3. More detail on the characterization of the neutralizing GPNMB antibody is required. There is virtually no information on the antibody provided. Where did it come from, how has it been validated, what assays demonstrate that it neutralizes GPNMB activity, what activity? Does the anti-GPNMB antibody alter fibroblast proliferation after seeding on the decellularized lung ECM?
4. CCK-8 assay data should be reported as cell number not viability.
5. Are the changes in migration in figures 3E due to an increase in cell proliferation?

6. Fig 4C. the resolution of the spatial transcriptomics is too low to really appreciate the cool data that is shown. This data is a strength of the manuscript and should be highlighted to exploit it better.

7. A western blot of GPNMB should be provided in Fig 6A to substantiate the IF shown

Text challenges

Line 71: 'fiber proliferation' is an odd word choice. I am not sure that is what the authors meant.

Line 171: 'toned' is misplaced, does not fit.

Line 174: delete 'to'

Line 185: change 'is' to 'to'

There are text issues with the referencing of Fig. 3G, H, I.

There are text issues referencing Fig. 6A and Fig. 7.

In general the text should be edited for clarity.

Minor:

- Figure 2A-C. Color scheme should be consistent. In panel A red is down in the other panels red is up.
- Quantification of 1K and 2H would be useful
- Fig 3A, please clarify how much GPNMB is added to the cells, please provide this in the legend.
- Fig. 7K, please indicate in the legend how much TGF β is added to the cells.

Reviewer #2 (Remarks to the Author):

In this manuscript, Wang et al show that the protein GPNMB is deposited on extra-cellular matrix during pulmonary fibrosis induced by silica. GPNMB exerts effects on isolated fibroblasts that are consistent with induction of a pro-fibrotic phenotype (e.g. increased survival of fibroblasts). Through proteomic and transcriptomic analyses, they build a case in which macrophage production of GPNMB "decorates" ECM, inducing pro-fibrotic changes in fibroblasts. The role of macrophages is supported by data showing that clodronate depletion of macrophages results in reduced production of GPNMB and a reduction in fibrosis. Macrophage- specific GPNMB expression is shown by scRNA sequencing and immunofluorescence. The authors also confirm GPNMB expression in murine macrophage cell lines Raw 264.7 and human monocyte cell line THP-1 upon SiO₂ treatment. Authors also show increased expression of GPNMB in patients with pulmonary fibrosis. While the silica model of fibrosis is beautiful and the work on GPNMB is quite nice, there are some major limitations with aspects of the manuscript. Many of the microscopic images are too small or too low-resolution to adequately judge the authors' claim. ("Zooming" in on the pdf did not help, as the images became pixelated.) The data regarding serpinB2 and CD44 are correlative and inconclusive. There is insufficient explanation of the human samples. Major and minor issues are listed below:

Major Points:

1. Many of the microscopic/histology images were too small or too low-resolution to appreciate the findings the authors wished to show. When "zoomed in," the image was too low resolution. Without increased magnification and resolution, it is difficult to be convinced of the differences the authors describe. This is a critical issue to address for the validity of the manuscript. Please provide higher magnification and higher resolution of images from Fig. 1A, Fig. S1A-C, Fig. 4A, 4C, Fig. S5B, Fig. 6A.
2. Legends of figures should provide more detail on the data shown in the graphs. For instance, in Fig. 1D, what does each symbol show? How many replicate experiments were performed? Is n the number of animals, or number of independent experiments?
3. In Fig. 1E, the authors state that Young's elastic modulus is increased in the silica-exposed lungs. There is no statistical difference in control and treated lungs. The authors cannot say that there is an

increase.

4. Line 219: There may be a slight increase in macrophage populations in SiO₂-7 groups but "obviously" is a bit strong. Also, is there some way of statistically testing whether the differences in cell populations is meaningful? There is no real increase in macrophages in the SiO₂-56 group, and the authors should discuss.
5. Fig. 4B/line 222: Authors state that GPNMB was "mainly derived" from macrophages based on scRNA-seq analysis, but it appears that AT-2 Igha+ cells also express GPNMB as well, especially in the SiO₂-56 group. Authors should address the possibility that GPNMB is also made by AT-2 cells, at least in the discussion.
6. Fig. 5A is insufficient evidence of macrophage depletion. Authors should quantify multiple images, at the very least.
7. Demonstration of GPNMB expression following silica treatment of the macrophage cell lines nicely shows that macrophages can produce GPNMB. The data would be strengthened if authors performed a similar experiment with isolated murine alveolar macrophages.
8. Posing CD44 as a molecular mechanism does not quite make sense, as CD44 activation deactivates Nf-κB signaling in astrocytes (PMID: 29519253) and also on macrophages to inhibit inflammatory response (J Immunol. 2007 May 15;178(10):6557-66, Cell Immunol. 2017 Jun;316:53-60). Analyzing the inflammatory response (cytokine or transcription factors) in fibroblasts in response to GPNMB would better demonstrate a potential pro-fibrotic effect in SiO₂ induced fibrosis.
9. To better demonstrate the proposed lung fibrosis progression through GPNMB \diamond CD44-Serpinb2 axis, author should show more of a molecular mechanism (downstream signaling cascade/molecular regulators) how this axis changes fibroblast trans differentiation during fibrosis progression?
10. The statement that "CD44/Serpinb2 is related to changes in fibroblast function driven by increased GPNMB expression..." is not supported by the data presented. Overall the CD44/serpinb2 results are speculative, correlative, and unhelpful to the overall paper.
11. Much more on the human samples is required. How were subjects recruited? How was PF clinically defined? Is it sampled from bronchoalveolar lavage fluid or lung biopsy? How was it quantified? Who approved the human studies protocol? Were the subjects with PF alive? Are these autopsy samples? I can find nothing on human subjects in the methods sections. (If it is human studies exempt, that should be stated.) The paucity of information on the human subjects is really inadequate.

Minor points

1. Please add scale bar to all panels/images (Fig. 1 A, 1D, 1K and other figures with images).
2. Please quantification fibroblast migration in Fig. 1K.
3. Fig. 1F: Why are the data normalized to 100%? The normalization should be justified, or raw data (non-normalized) should be shown.
4. Add immunoblot for loading control in Fig. 2G.
5. Line 165: Authors state that z-SMA is increased, but the y-axis says ACTA2. Are they the same thing, or is one label incorrect?
6. Line 166: Fig. 1I is not mentioned.
7. Lines 171-174: These sentences do not make sense. "Toned?" Please rewrite.
8. Line 180: Other proteins appear to be upregulated as highly as GPNMB. Would therefore describe it as "one of the most highly upregulated..." not, "the most highly upregulated..."
9. Lines 184-185: Please provide references for the Kegg and Gene Oncology data. Presumably these were performed using online tools, there should be a reference.
10. All the data pertaining to Figure 3: Please clarify – preferably with a short sentence in the Results section – how the fibroblasts were exposed to GPNMB. Presumably in vitro? But this is not clear.
11. Line 235: It's very hard to "prove" anything in science. Would suggest using a word like "To test if the cellular origin..." or "To investigate the cellular origin..."
12. What are "macrophage scavengers" in line 238 and 240? Clodronate? SiO₂? The term "macrophage scavengers" is confusing.
13. Line 259: Not Figure 7A, but Fig. 6A.
14. Add density measurement on immunoblots in Fig. 6B, 6C, Fig. 7E, 7K and S6D.

15. Line 272: Please clarify how the experiment was done. Were RAW cells exposed to silica, and THEN transferred to decellularized ECM? Or was ECM repopulated with RAW cells, and then exposed to silica? It's confusing as written, and matters for interpretation. It seems the best way to do the experiment would be to treat the RAW cells with silica in vitro, then transfer them to decellularized ECM.

16. Fig. S7 is incorrectly labeled at the top as Fig. S6.

Statistical analysis: t-test used throughout. As long as data are distributed normally, this is okay. If data is not distributed normally, a Mann-Whitney should be used. As indicated, authors need to provide more information about animal, sample number and number of experimental replicates.

Reviewer #3 (Remarks to the Author):

Wang/Zhang/Yuan et al., have identified investigated the contributions of the extracellular matrix (ECM) to pulmonary fibrosis in mice using a silica induced model of pulmonary fibrosis. The authors have identified GPNMB as a driver of fibrotic changes to fibroblasts within the ECM. They propose that macrophages are the principal cellular source of GPNMB.

The use of complimentary multi-omics approaches (e.g., scRNAseq, proteomics and spatial transcriptomics) to confirm their findings at both the gene and protein level is strength. The authors have also investigated the expression of GPNMB in a cohort of patients with pulmonary fibrosis (IPF) making the study clinically relevant.

While this works presents interesting and novel findings regarding the role of the ECM on lung fibroblast function, it would benefit from some additional revision. The authors use vague terms and obscure language throughout the manuscript, this makes the manuscript confusing and often hard to follow. This should be corrected in order to improve clarity for the reader. Specific examples include: 'has an effect on', 'alleviates the effect of', can the authors simply state if they believe the changes observed to be pro- or anti- fibrotic?

It is difficult to assess the robustness of the findings presented within the manuscript as some data has been omitted e.g., complete list of DEGs and DE proteins. The statistical tests performed to analyze the data are not clearly explained (multiple t-tests), if this is the case they should be corrected for multiple comparisons and the distribution of the data should be assessed e.g., normal distribution.

Major concerns:

1. In Figure 1 the authors state that co-culture with fibrotic ECM causes and increase in cellular viability (CCK-8) assay in normal lung fibroblasts. The authors need to clearly define what this assay measures e.g., metabolic activity, cell stress etc., as later in the manuscript (Fig 3A-C) the authors state that 'increased viability' is due to elevated cell number and proliferation of fibroblasts.

2. The manuscript assumes that changes observed in lung fibroblasts are specific to matrix derived GPNMB, however the ECM is reservoir of numerous growth factors (TGFB, PDGF, FGF). The authors should explain this in more detail or perform additional experiments (as per Figure 3) to show these changes are specific to GPNMB.

3. The volcano plots in Figs 2C and 7B list several differentially expressed proteins and genes in fibroblast following exposure to 'normal' and fibrotic ECM. However, only GPMNB is listed. This makes the validity of the findings and network analysis difficult to interpret and the names of at least the top-ranking genes, and proteins should be included.

4. While RAW264.7 are a murine macrophage cell line, THP-1 cells are human monocyte cell line that need to be differentiated into macrophages e.g., via PMA treatment. Can the authors please explain how this human monocyte cell line is relevant to their murine studies? They should also verify the changes they observe following silica treatment (Figure 6D) using primary murine lung macrophages.

5. No consideration is given to fibroblast-macrophage crosstalk, due to the considerable GPNMB gene expression by lung fibroblasts in the transcriptome studies (Figure 4B) this should be discussed.

6. All statistical tests should be listed in the figure legends, the rationale for selecting them should be clearly explained e.g. sample size, distribution, corrections for multiple comparisons etc. This will ensure the validity and robustness of the findings presented.

Minor concerns:

1. scRNAseq figure 4A missing cell subset labels 11-15

2. Are the labels (GSE183683) and (PD PXD028194) listed in supplementary information linked to publicly available repositories? If this is not the case, what is this terminology.

3. More information should be provided on the mouse lung fibroblast cell line used in the study e.g., cell line name/ID, baseline characteristics, passage number used etc.

Date:

To: reviewers, Communications Biology

Re: Decision on “Macrophage-derived GPNMB trapped by fibrotic extracellular matrix promotes pulmonary fibrosis”

Response to Critiques

We would like to thank the the reviewers for their positive and constructive comments on our manuscript. We are submitting the attached revised manuscript, in which the reviewer’s recommendations have been incorporated, for your consideration. A summary of the revisions and a point-by-point response to the critiques are provided below. Considering the contribution of Min Long to the article during the revision process, we added Min Long as the co-author.

Additionally, we would like to express our appreciation to *Communications Biology* for helping us to improve this manuscript.

Reviewers' comments:

Reviewer #1 (Remarks to the Author):

Wang et al investigate the contribution of fibrotic ECM on fibroblast gene expression and phenotype in context of an experimental model of fibrosis. A

major theme of the study is the development of an interesting model system to address how ECM associated with pulmonary fibrosis impacts cell fate. Pulmonary fibrosis is a significant unmet need worldwide. Improved model systems to study disease pathogenesis and the biology of cells associated with pulmonary fibrosis are needed. Here the authors instigate pulmonary fibrosis by intratracheal instillation of silica into mice. Lung fibrosis is evaluated at day 7 and day 56 post silica administration. The authors demonstrate through lung functional assays that silica administration does indeed reduce lung function in the mice. Lung fibrotic tissue is used for downstream analysis. Interestingly lung tissue is also subjected to decellularization and subsequently seeded by mouse fibroblasts to investigate how the ECM of normal lung tissue compares to ECM of fibrotic lung tissue in terms of fibroblast gene/protein expression and phenotype. This is a strength of the study. The authors perform proteomic and transcriptomic assays leading to the identification of GPNMB as potentially involved in the fibrotic phenotype post silica administration. After cell based and expression assays the authors conclude that macrophage derived GPNMB promotes lung fibrosis. Overall the study is interesting, exploits useful approaches and has potential to provide impact. Concerns are mainly centered around interpretation of the data and inflation of the strength of the data to draw conclusions. Specific comments are below.

Response: We sincerely thank you for the positive comments.

Major Comments:

1. A major concern is the tendency of the authors to over-interpret the data leading to some conclusions are not supported. Some examples are below

- Fig 1E: the data do not indicate a change in stiffness in NS vs SiO₂ treated lungs yet the authors on line 156 indicate that stiffness is elevated in lungs from SiO₂ treated mice.

Response: We apologize for this oversight. The stiffness in NS- vs. SiO₂-treated lungs showed an increasing trend, but there was no significant difference (due to the limitation of the sample size). Based on your suggestion, the text description has been modified in the article.

- Fig 2C: the authors state on Line 180 that GPNMB is the most highly upregulated protein in PF. This is not true according to the data presented in Fig. 2C.

Response: We apologize for the misunderstanding caused by an inappropriate description. GPNMB was identified as the most highly upregulated protein that may be related to the pulmonary fibrosis process in silica-treated mice. We have modified the description in the article. Thank you for your suggestion.

- Fig 4A: the authors state in lines 223-224 that GPNMB was obviously increased in SiO₂-56 day 56 group mice, but this is not clear at all based on the

data shown in Fig. 4A.

Response:

Response: We apologize for this oversight. A violin chart is shown in Fig S3J to display the changes in GPNMB in the different groups. Thank you for your suggestion.

- Fig 4: in general the authors discount the potential contribution of other cell types that express GPNMB.

Response: We apologize for this oversight. GPNMB is a membrane-bound glycoprotein that is expressed in many cell types, such as osteoclasts, osteoblasts, immune cells, melanocytes, and epithelial cells. The widespread expression of GPNMB indicates the involvement of GPNMB in many physiological processes. In our study, we mainly focused on the contribution of macrophage-derived GPNMB to the pulmonary fibrosis process while not excluding the contribution of GPNMB from other cell types in this process.

- Fig 7: the authors mine the omics data to craft a potential pathway that fits the change in GPNMB expression. There is no functional data to support that Serpinb2 or CD44 are functionally involved based on the data provided. The authors should soft pedal their conclusions. The data shown in Fig. 7B-K are modest at best.

None of these are fatal flaws. The data are useful and the authors, in my opinion,

should downplay hard conclusions. Their data are very nice but do not provide meaningful functional demonstration of the contribution of GPNMB or Serpinb2/CD44 to lung fibrosis. The data are descriptive and that is ok and useful for the field.

Response: We apologize for this oversight. We performed other experiments according to your suggestion. siRNA-induced CD44 knockdown in fibroblasts suppressed cell migration induced by GPNMB treatment (Fig S8A), thus suggesting the contribution of GPNMB/CD44/Serpinb2 to lung fibrosis. Thank you for your valuable advice.

2. More detail is required on how the 3D migration assay is performed and quantified (Fig 1K, 3H, 5F).

Response: We apologize for this oversight. Thank you for your valuable advice. We have added the method of the 3D migration assay to the methods section.

3. More detail on the characterization of the neutralizing GPNMB antibody is required. There is virtually no information on the antibody provided. Where did it come from, how has it been validated, what assays demonstrate that it neutralizes GPNMB activity, what activity? Does the anti-GPNMB antibody alter fibroblast proliferation after seeding on the decellularized lung ECM?

Response: We apologize for this oversight. The neutralizing GPNMB antibody (R&D Systems) was bought to observe the effects of GPNMB in ECM, and we

have stated this information in the article. Our data showed that the neutralizing GPNMB antibody (0.5 µg/mL) mitigated fibroblast migration induced by fibrotic ECM, thus suggesting a contribution of GPNMB to fibroblast migration. Based on our experimental results, we speculated that GPNMB had an impact on fibroblast proliferation, and we would discuss it in the subsequent experiments. Thank you for your valuable suggestions.

4. CCK-8 assay data should be reported as cell number not viability.

Response: We apologize for this oversight. We have shown CCK-8 assay data as the cell number (reflected by the OD values of the cells according to papers reported) according to your advice. Thank you for your advice.

5. Are the changes in migration in figures 3E due to an increase in cell proliferation?

Response: According to Figure 3E, cell migration increased after GPNMB treatment. However, we cannot exclude the phenomenon of cell proliferation after GPNMB treatment; indeed, GPNMB also affected cell proliferation (Figures 3 C and D).

6. Fig 4C. the resolution of the spatial transcriptomics is too low to really appreciate the cool data that is shown. This data is a strength of the manuscript and should be highlighted to exploit it better.

Response: We apologize for this oversight. We have provided high-resolution images in the text and in the supplementary images file. Thank you for your valuable advice.

7. A western blot of GPNMB should be provided in Fig 6A to substantiate the IF shown

Response: We apologize for this oversight. We have supplied a western blot of GPNMB to substantiate the IF shown (Fig S5F). Thank you for your valuable advice.

Text challenges

Line 71: 'fiber proliferation' is an odd word choice. I am not sure that is what the authors meant.

Response: We apologize for this oversight, and we have replaced the term 'fibroproliferation' with 'fiber proliferation'. Thank you for your valuable advice.

Line 171: 'toned' is misplaced, does not fit.

Response: We apologize for this oversight, and we have replaced the phrase 'were needed' with 'toned'. Thank you for your valuable advice.

Line 174: delete 'to'

Response: We apologize for this oversight, and we have deleted the term 'to'.

Thank you for your valuable advice.

Line 185: change 'is' to 'to'

Response: We apologize for this oversight, and we have changed the term 'is' to 'to'. Thank you for your valuable advice.

There are text issues with the referencing of Fig. 3G, H, I.

Response : We apologize for this oversight, and we have improved the referencing of Fig. 3G, H, I in the figure titles and legends. Thank you for your valuable advice.

There are text issues referencing Fig. 6A and Fig. 7.

Response : We apologize for this oversight, and we have improved the referencing of Fig. 6A and Fig. 7 in the figures and figure titles and legends. Thank you for your valuable advice.

In general the text should be edited for clarity.

Response: Thank you for your valuable advice. The text has been edited for clarity according to your suggestion.

Minor:

- Figure 2A-C. Color scheme should be consistent. In panel A red is down in the other panels red is up.

Response: We apologize for this oversight, and we have adjusted the image colors. Thank you for your valuable advice.

- Quantification of 1K and 2H would be useful

Response: Thank you for your valuable advice. Quantification of 1K is shown in Fig S2C and D.

- Fig 3A, please clarify how much GPNMB is added to the cells, please provide this in the legend.

Response: We apologize for this oversight, and we have provided the dose of GPNMB that was added to the cells in the legend. Thank you for your valuable advice.

- Fig. 7K, please indicate in the legend how much TGF β is added to the cells.

Response: We apologize for this oversight, and we have provided the dose of TGF β that was added to the cells in the legend. Thank you for your valuable advice.

Reviewer #2 (Remarks to the Author):

In this manuscript, Wang et al show that the protein GPNMB is deposited on extra-cellular matrix during pulmonary fibrosis induced by silica. GPNMB exerts effects on isolated fibroblasts that are consistent with induction of a pro-fibrotic phenotype (e.g. increased survival of fibroblasts). Through proteomic and transcriptomic analyses, they build a case in which macrophage production of GPNMB “decorates” ECM, inducing pro-fibrotic changes in fibroblasts. The role of macrophages is supported by data showing that clodronate depletion of macrophages results in reduced production of GPNMB and a reduction in fibrosis. Macrophage- specific GPNMB expression is shown by scRNA sequencing and immunofluorescence. The authors also confirm GPNMB expression in murine macrophage cell lines Raw 264.7 and human monocyte cell line THP-1 upon SiO₂ treatment. Authors also show increased expression of GPNMB in patients with pulmonary fibrosis. While the silica model of fibrosis is beautiful and the work on GPNMB is quite nice, there are some major limitations with aspects of the manuscript. Many of the microscopic images are too small or too low-resolution to adequately judge the authors’ claim. (“Zooming” in on the pdf did not help, as the images became pixelated.) The data regarding serpinB2 and CD44 are correlative and inconclusive. There is insufficient explanation of the human samples. Major and minor issues are listed below:

Response: We sincerely appreciate your positive comments.

Major Points:

1. Many of the microscopic/histology images were too small or too low-resolution to appreciate the findings the authors wished to show. When “zoomed in,” the image was too low resolution. Without increased magnification and resolution, it is difficult to be convinced of the differences the authors describe. This is a critical issue to address for the validity of the manuscript. Please provide higher magnification and higher resolution of images from Fig. 1A, Fig. S1A-C, Fig. 4A, 4C, Fig. S5B, Fig. 6A.

Response: We apologize for this oversight. We have provided high-resolution images in the text and in the supplementary images file. Thank you for your valuable advice.

2. Legends of figures should provide more detail on the data shown in the graphs. For instance, in Fig. 1D, what does each symbol show? How many replicate experiments were performed? Is n the number of animals, or number of independent experiments?

Response: We apologize for this oversight. More detail has been provided in the figure legends according to your advice. Five replicate experiments were performed, and n is the number of animals. Thank you for your valuable advice.

3. In Fig. 1E, the authors state that Young’s elastic modulus is increased in the

silica-exposed lungs. There is no statistical difference in control and treated lungs. The authors cannot say that there is an increase.

Response: We apologize for this oversight. The text description has been modified in the article.

Although the Young's elastic modulus in the silica-exposed lungs showed a certain growth trend, there was no significant difference between the control and treated lungs. There were only 3 samples in each group for Young's elastic modulus, and increasing the sample size may be needed to evaluate the stiffness of the lungs. We apologize for the inappropriate description.

4. Line 219: There may be a slight increase in macrophage populations in SiO₂-7 groups but "obviously" is a bit strong. Also, is there some way of statistically testing whether the differences in cell populations is meaningful? There is no real increase in macrophages in the SiO₂-56 group, and the authors should discuss.

Response: We apologize for this oversight. We have improved our description according to your advice. A violin chart is shown in Fig S3J to display the changes in GPNMB in the different groups. The data indicate that the inflammatory reaction usually occurs in the early stage after silica treatment; thus, there was no further obvious increase in macrophages in the SiO₂-56 group than in the SiO₂-7 group. However, the macrophages in the SiO₂-56 group increased compared with those in NS-56. Thank you for your valuable

advice.

5. Fig. 4B/line 222: Authors state that GPNMB was “mainly derived” from macrophages based on scRNA-seq analysis, but it appears that AT-2 Igha+ cells also express GPNMB as well, especially in the SiO₂-56 group. Authors should address the possibility that GPNMB is also made by AT-2 cells, at least in the discussion.

Response: We apologize for this oversight. We have discussed the origin of GPNMB according to your suggestion. Thank you for your valuable advice.

‘Furthermore, scRNA-seq also showed increased GPNMB mRNA levels in other cells after silica instillation, such as AT-2 cells, and we mainly focused on macrophage-derived GPNMB according to our included data. However, our study could not exclude the fact that GPNMB from other cells, which may also be involved in the process of pulmonary fibrosis induced by silica.’

6. Fig. 5A is insufficient evidence of macrophage depletion. Authors should quantify multiple images, at the very least.

Response : We apologize for this oversight. We administered clodronate liposomes to mice every 7 days to remove macrophages. We have quantified the images, and the data are shown in Fig S5E. Thank you for your valuable advice.

7. Demonstration of GPNMB expression following silica treatment of the macrophage cell lines nicely shows that macrophages can produce GPNMB. The data would be strengthened if authors performed a similar experiment with isolated murine alveolar macrophages.

Response: Thank you for your valuable advice. We performed a similar experiment with isolated murine macrophages, and the data are shown in Fig S6C and D.

8. Posing CD44 as a molecular mechanism does not quite make sense, as CD44 activation deactivates Nf-kB signaling in astrocytes (PMID: 29519253) and also on macrophages to inhibit inflammatory response (J Immunol. 2007 May 15;178(10):6557-66, Cell Immunol. 2017 Jun;316:53-60). Analyzing the inflammatory response (cytokine or transcription factors) in fibroblasts in response to GPNMB would better demonstrate a potential pro-fibrotic effect in SiO₂ induced fibrosis.

Response: Thank you very much. CD44 activation plays a role in inflammatory response. However, In our study, the inflammatory response of fibroblasts cultured in fibrotic ECM was not showed obvious change, as represented by the transcriptome analysis. Therefore, we supposed GPNMB might not affect cell behaviors via inflammatory response in our study. Thank you for your valuable advice.

9. To better demonstrate the proposed lung fibrosis progression through GPNMB \diamond CD44-Serpinb2 axis, author should show more of a molecular mechanism (downstream signaling cascade/molecular regulators) how this axis changes fibroblast trans differentiation during fibrosis progression?

Response : Thank you for your valuable advice. We supplemented the experiment according to your suggestion. siRNA-induced CD44 knockdown in fibroblasts suppressed cell migration induced by GPNMB treatment (Fig S8A), thus suggesting the contribution of this axis to lung fibrosis. Moreover, the immunofluorescence staining data of lung tissues showed that CD44 and Serpinb2 levels increased 56 days after silica instillation (Fig S9A and B).

10. The statement that “CD44/Serpinb2 is related to changes in fibroblast function driven by increased GPNMB expression...” is not supported by the data presented. Overall the CD44/serpinb2 results are speculative, correlative, and unhelpful to the overall paper.

Response: We apologize for this oversight. We speculated that the changes in fibroblast function caused by GPNMB may be related to CD44, according to the omics results and literature research. siRNA-induced CD44 knockdown in fibroblasts suppressed cell migration induced by GPNMB treatment (Fig S8A), thus suggesting the contribution of GPNMB/CD44/Serpinb2 to lung fibrosis. Thank you for your valuable advice.

11. Much more on the human samples is required. How were subjects recruited? How was PF clinically defined? Is it sampled from bronchoalveolar lavage fluid or lung biopsy? How was it quantified? Who approved the human studies protocol? Were the subjects with PF alive? Are these autopsy samples? I can find nothing on human subjects in the methods sections. (If it is human studies exempt, that should be stated.) The paucity of information on the human subjects is really inadequate.

Response: We apologize for the unclear description. Data for human samples were obtained from GEO datasets.

Minor points

1. Please add scale bar to all panels/images (Fig. 1 A, 1D, 1K and other figures with images).

Response: We apologize for this oversight. We have added a scale bar to all of the panels/images according to your advice. Thank you for your valuable advice.

2. Please quantification fibroblast migration in Fig. 1K.

Response: We apologize for the unclear description. The quantification of fibroblast migration is shown in Fig S2C and D.

3. Fig. 1F: Why are the data normalized to 100%? The normalization should be justified, or raw data (non-normalized) should be shown.

Response: We apologize for this oversight. We have shown the raw data (nonnormalized) according to your advice. Thank you for your valuable advice.

4. Add immunoblot for loading control in Fig. 2G.

Response: We apologize for the unclear description. Extracellular matrix proteins maintain a dynamic change process in vivo. They do not have the same housekeeping proteins as cellular proteins; thus, there is no internal parameter. In this experiment, we extracted proteins according to different tissue qualities and then quantified them to ensure that the same amount of protein was added to each well.

5. Line 165: Authors state that α -SMA is increased, but the y-axis says ACTA2. Are they the same thing, or is one label incorrect?

Response: We apologize for this oversight. ACTA2 is the gene name of α -SMA, and we have modified the description in the article. Thank you very much.

6. Line 166: Fig. 1I is not mentioned.

Response: We apologize for this oversight. We have modified this information in the article. Thank you very much for your suggestion.

7. Lines 171-174: These sentences do not make sense. "Toned?" Please rewrite.

Response: We apologize for this oversight. We have modified this information in the article. Thank you very much for your suggestion.

8. Line 180: Other proteins appear to be upregulated as highly as GPNMB. Would therefore describe it as "one of the most highly upregulated..." not, "the most highly upregulated..."

Response: We apologize for this oversight. We have changed the description according to your advice. Thank you very much for your suggestion.

9. Lines 184-185: Please provide references for the Kegg and Gene Oncology data. Presumably these were performed using online tools, there should be a reference.

Response: We apologize for this oversight. We have provided references for the Kegg and Gene Oncology data in the Methods section according to your advice (references 40, 41, 42). Thank you for your valuable advice.

40. Kanehisa M, Goto S. KEGG: kyoto encyclopedia of genes and genomes. Nucleic Acids Res 2000; 28: 27-30.

41. Ashburner et al. Gene ontology: tool for the unification of biology. Nat Genet 2000; 25: 25-9.

42. Gene Ontology Consortium. The Gene Ontology resource: enriching a GOLD mine. *Nucleic Acids Res* 2021; 49: D325-D334.

10. All the data pertaining to Figure 3: Please clarify – preferably with a short sentence in the Results section – how the fibroblasts were exposed to GPNMB. Presumably in vitro? But this is not clear.

Response: We apologize for this oversight. The fibroblasts were exposed to GPNMB in vitro. We have changed the description according to your advice. Thank you for your valuable advice.

11. Line 235: It's very hard to “prove” anything in science. Would suggest using a word like “To test if the cellular origin...” or “To investigate the cellular origin...”

Response: We apologize for this oversight. We have changed the description according to your advice. Thank you for your valuable advice.

12. What are “macrophage scavengers” in line 238 and 240? Clodronate? SiO₂?

The term “macrophage scavengers” is confusing.

Response: We apologize for this oversight. Clodronate liposomes, which are taken up by macrophages via phagocytosis, can strongly reduce the number of macrophages. The macrophage scavengers are clodronate liposomes. We have changed the description in the article to avoid confusion. Thank you for your valuable advice.

13. Line 259: Not Figure 7A, but Fig. 6A.

Response: We apologize for this oversight. We have changed the description according to your reminder. Thank you for your valuable advice.

14. Add density measurement on immunoblots in Fig. 6B, 6C, Fig. 7E, 7K and S6D.

Response : We apologize for the unclear description. The density measurements on the immunoblots are shown in the supplementary figure.

Figure. S6A for Fig. 6B

Figure S6B for Fig. 6C

Figure S6E for Fig. 7F(Fig 7K before the revisions)

Figure S6H for Fig. 7I (Fig 7K before the revisions)

Figure S6G (S6D before the revisions) for Fig. S6F

Thank you very much for your suggestions.

15. Line 272: Please clarify how the experiment was done. Were RAW cells exposed to silica, and THEN transferred to decellularized ECM? Or was ECM repopulated with RAW cells, and then exposed to silica? It's confusing as written and matters for interpretation. It seems the best way to do the experiment would be to treat the RAW cells with silica in vitro, then transfer them to decellularized ECM.

Response: We apologize for the unclear description. In our experiment, ECM repopulated RAW264.7 cells, which were then exposed to silica. In our opinion, it is more reasonable to design the experiment in this manner. If RAW cells are transferred to decellularized ECM after silica treatment in vitro, it may not be possible to observe a convincing phenomenon because GPNMB may be produced before being transferred to ECM. To better explain the experimental process, we have added an experimental flow chart in Fig. S7A. Thank you very much for your suggestion.

16. Fig. S7 is incorrectly labeled at the top as Fig. S6.

Response: We apologize for this oversight. We have changed the description according to your reminder. Thank you very much for your suggestion.

Statistical analysis: t-test used throughout. As long as data are distributed normally, this is okay. If data is not distributed normally, a Mann-Whitney should be used. As indicated, authors need to provide more information about animal, sample number and number of experimental replicates.

Response: We apologize for this oversight. We have changed the description according to your suggestion. All of the data are presented as the mean \pm standard error of the mean (SEM), with sample sizes and numbers of repeats indicated in the figure legends. Comparisons between the two groups were analyzed by using a two-tailed Student's t test. One-way ANOVA with the

Bonferroni test for multiple comparisons was used to compare multiple groups.

Thank you for your valuable advice.

Reviewer #3 (Remarks to the Author):

Wang/Zhang/Yuan et al., have identified investigated the contributions of the extracellular matrix (ECM) to pulmonary fibrosis in mice using a silica induced model of pulmonary fibrosis. The authors have identified GPNMB as a driver of fibrotic changes to fibroblasts within the ECM. They propose that macrophages are the principal cellular source of GPNMB.

The use of complimentary multi-omics approaches (e.g., scRNAseq, proteomics and spatial transcriptomics) to confirm their findings at both the gene and protein level is strength. The authors have also investigated the expression of GPNMB in a cohort of patients with pulmonary fibrosis (IPF) making the study clinically relevant.

While this works presents interesting and novel findings regarding the role of the ECM on lung fibroblast function, it would benefit from some additional revision. The authors use vague terms and obscure language throughout the manuscript, this makes the manuscript confusing and often hard to follow. This should be corrected in order to improve clarity for the reader. Specific examples

include: 'has an effect on', 'alleviates the effect of', can the authors simply state if they believe the changes observed to be pro- or anti- fibrotic?

It is difficult to assess the robustness of the findings presented within the manuscript as some data has been omitted e.g., complete list of DEGs and DE proteins. The statistical tests performed to analyze the data are not clearly explained (multiple t-tests), if this is the case they should be corrected for multiple comparisons and the distribution of the data should be assessed e.g., normal distribution.

Response: We greatly appreciate your positive comments.

Major concerns:

1. In Figure 1 the authors state that co-culture with fibrotic ECM causes and increase in cellular viability (CCK-8) assay in normal lung fibroblasts. The authors need to clearly define what this assay measures e.g., metabolic activity, cell stress etc., as later in the manuscript (Fig 3A-C) the authors state that 'increased viability' is due to elevated cell number and proliferation of fibroblasts.

Response: We apologize for this oversight. We have changed the description according to your suggestion. The Cell Counting Kit-8 (CCK-8) assay was used to evaluate cell viability (the proliferation and growth of cells) in different ECMs.

Thank you very much for your valuable advice.

2. The manuscript assumes that changes observed in lung fibroblasts are specific to matrix derived GPNMB, however the ECM is reservoir of numerous growth factors (TGFB, PDGF, FGF). The authors should explain this in more detail or perform additional experiments (as per Figure 3) to show these changes are specific to GPNMB.

Response: Thank you very much for your advice. For the proteomic analysis of the ECM, we analyzed the data for the candidate proteins by using a Student's t test to identify proteins that showed significant differences in expression between different groups. The fold change (FC) was used to evaluate the expression levels of individual proteins between samples. The p value was calculated by using the t test to determine the significance of the difference between the samples. The screening conditions were $FC \geq 2.0$ and $P \leq 0.05$. The ECM is a reservoir of numerous growth factors, including TGF β 1, PDGF, and FGF, and growth factors play important roles in cell behavior. However, the data showed that the changes in TGF β 1, PDGF, and FGF were not significant ($FC \geq 2.0$ and $P \leq 0.05$), whereas we focused on the role of proteins that changed significantly between different groups ($FC \geq 2.0$ and $P \leq 0.05$) in cell functions.

3. The volcano plots in Figs 2C and 7B list several differentially expressed proteins and genes in fibroblast following exposure to 'normal' and fibrotic ECM. However, only GPMNB is listed. This makes the validity of the findings and

network analysis difficult to interpret and the names of at least the top-ranking genes, and proteins should be included.

Response: We apologize for this oversight. We have listed the top-ranking proteins in Figure S3 H and Table S1. For the transcriptomic analysis, we have listed the top-ranking proteins in Table S2. Thank you very much for your valuable advice.

4. While RAW264.7 are a murine macrophage cell line, THP-1 cells are human monocyte cell line that need to be differentiated into macrophages e.g., via PMA treatment. Can the authors please explain how this human monocyte cell line is relevant to their murine studies? They should also verify the changes they observe following silica treatment (Figure 6D) using primary murine lung macrophages.

Response: Thank you very much for your advice. In our experiment, THP-1 cells were differentiated into macrophages via PMA treatment. We obtained primary murine macrophages and then observed changes in GPNMB protein levels after silica treatment. The data showed increased GPNMB protein levels in macrophages after silica treatment (Figure S6 C-D).

5. No consideration is given to fibroblast-macrophage crosstalk, due to the considerable GPNMB gene expression by lung fibroblasts in the transcriptome

studies (Figure 4B) this should be discussed.

Response: Thank you very much for your advice. In our study, we did not exclude fibroblast-macrophage crosstalk after silica treatment. Studies have found that macrophages in the lung increase inconspicuously in the fibrosis stage. However, proteins released from macrophages continue to accumulate and may affect fibroblast behavior.

6. All statistical tests should be listed in the figure legends, the rationale for selecting them should be clearly explained e.g. sample size, distribution, corrections for multiple comparisons etc. This will ensure the validity and robustness of the findings presented.

Response: We apologize for this oversight. We have changed the description according to your suggestion. Thank you very much for your suggestions.

'All of the data are presented as the mean \pm standard error of the mean (SEM), with sample sizes and numbers of repeats indicated in the figure legends. Comparisons between the two groups were analyzed using a two-tailed Student's t test. One-way ANOVA with the Bonferroni test for multiple comparisons was used to compare multiple groups. Differences were considered significant at $p < 0.05$. All analyses were conducted using GraphPad Prism version 8.0.

Minor concerns:

1. scRNAseq figure 4A missing cell subset labels 11-15

Response: We apologize for this oversight. We have included the missing cell subset labels. Thank you very much for your suggestion.

2. Are the labels (GSE183683) and (PD PXD028194) listed in supplementary information linked to publicly available repositories? If this is not the case, what is this terminology.

Response: We apologize for the unclear description. Spatial transcriptomics (GSE183683) and proteomic analysis of the ECM (PD PXD028194) are data from our laboratory, and the data have been made public.

The RNA sequencing data have been deposited into the Gene Expression Omnibus Datasets through the National Center for Biotechnology Information (<https://www.ncbi.nlm.nih.gov/>) with the dataset identifier GSE183683.

The mass spectrometry proteomics data have been deposited to the ProteomeXchange Consortium (<http://proteomecentral.proteomexchange.org>) via the iProX partner repository [1] with the dataset identifier PXD028194.

[1] Ma J, et al. (2019) iProX: an integrated proteome resource. *Nucleic Acids Res*, 47, D1211-D1217.

3. More information should be provided on the mouse lung fibroblast cell line used in the study e.g., cell line name/ID, baseline characteristics, passage number used etc.

Response: We apologize for this oversight. We have provided more information on the mouse lung fibroblast cell line that was used in the study. Thank you very much for your valuable advice.

REVIEWERS' COMMENTS:

Reviewer #1 (Remarks to the Author):

The authors have provided an adequate response to the queries. It is an interesting study.

Reviewer #2 (Remarks to the Author):

The authors have responded to all concerns raised during the first review.

Reviewer #3 (Remarks to the Author):

The authors have addressed the majority of my concerns and I have no further comments to add.